# Towards Online Multimodal Social Interaction Understanding

**Xinpeng Li**                                                                          *xinpeng.li@utdallas.edu*
*University of Texas at Dallas*

**Shijian Deng**                                                                        *shijian.deng@utdallas.edu*
*University of Texas at Dallas*

**Bolin Lai**                                                                              *bolin.lai@gatech.edu*
*Georgia Institute of Technology*

**Weiguo Pian**                                                                          *weiguo.pan@utdallas.edu*
*University of Texas at Dallas*

**James Matthew Rehg**                                                                      *jrehg@illinois.edu*
*University of Illinois Urbana-Champaign*

**Yapeng Tian**                                                                          *yapeng.tian@utdalls.edu*
*University of Texas at Dallas*

**Reviewed on OpenReview:** *https://openreview.net/forum?id=5P7yVfUEuD*

## Abstract

In this paper, we introduce a new problem, Online-MMSI, where the model must perform multimodal social interaction understanding (MMSI) using only historical information. Given a recorded video and a multi-party dialogue, the AI assistant is required to immediately identify the speaker's referent, which is critical for real-world human-AI interaction. Without access to future conversational context, both humans and models experience substantial performance degradation when moving from offline to online settings. To tackle the challenges, we propose Online-MMSI-VLM, a novel framework based on multimodal large language models. The core innovations of our approach lie in two components: (1) multi-party conversation forecasting, which predicts upcoming speaker turns and utterances in a coarse-to-fine manner; and (2) socially-aware visual prompting, which highlights salient social cues in each video frame using bounding boxes and body keypoints. Our model achieves state-of-the-art results on three tasks across two datasets, significantly outperforming the baseline and demonstrating the effectiveness of Online-MMSI-VLM. Project page: https://sampson-lee.github.io/online-mmsi-project-page.

## 1 Introduction

Multimodal Social Interaction Understanding (MMSI) plays a critical role in advancing human-AI social interaction, aiming to interpret social behaviors by jointly leveraging verbal and non-verbal cues (Lee et al., 2024b; Lai et al., 2023; Lee et al., 2024a). Recent work has explored tasks such as speaking target identification, pronoun coreference resolution, and mentioned player prediction. One common goal is to identify a speaker's referent among multiple participants using recorded dialogues and video streams. For example, as illustrated in Figure 1 (1), when a user asks *Which player is the speaker referring to?* at a moment, the model analyzes the multimodal input and responds, "Player0." MMSI can enable applications such as AI-powered assistants for social support and collaborative tasks. Smart AR glasses, for instance, can help

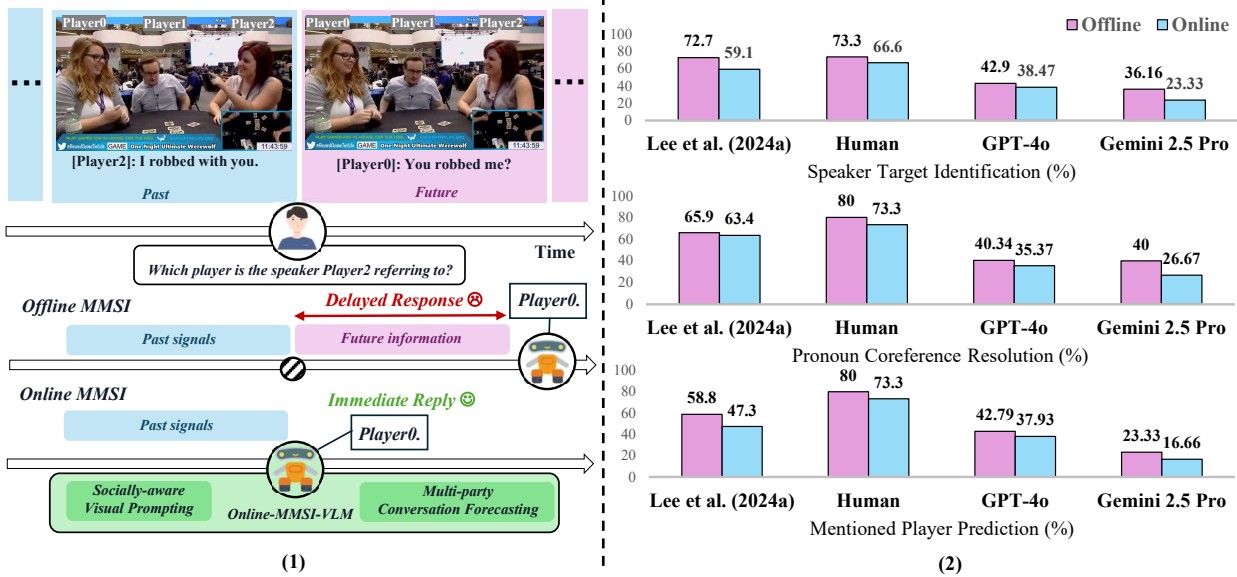

Figure 1: (1) Existing MMSI studies depend on both past and future context (offline setting), limiting their practicality in real-world scenarios that require immediate interpretation and response. To bridge the gap, we introduce a new problem, Online-MMSI, where the model must perform MMSI using only historical information (online setting). To address the challenge, we propose Online-MMSI-VLM, a novel multimodal large language model-based framework that integrates multi-party conversation forecasting and socially-aware visual prompting. (2) Performance drops significantly when both models and humans shift from offline to online settings across three social tasks on the YouTube dataset (Lai et al., 2023).

autistic individuals better understand social cues (Haber et al., 2020; Elsherbini et al., 2023), while intelligent assistants can actively participate in social settings (Breazeal et al., 2016). As a result, MMSI has attracted growing interest from the social AI research community (Lee et al., 2024b; Li et al., 2024a; Feng et al., 2025).

Prior MMSI studies focused on *offline* setting, conducting referent identification by leveraging both past and future context at a given moment (Lee et al., 2024a). They rely on extended transcripts and video data, producing responses with inherent delays. However, real-world AI assistants are expected to respond in real time, interpreting and responding to social dynamics without access to future information (i.e., *online* setting). To bridge this gap, we introduce Online-MMSI, a new problem formulation where models must perform MMSI tasks using only historical information. This setting is crucial for building responsive, real-time intelligent systems that are practical and deployable in real-world social environments.

Yet, Online-MMSI presents substantial and unique challenges compared with traditional offline settings. As shown in Figure 1 (2), performance drops significantly when both models and humans shift from offline to online settings across three social tasks. First, online models lack access to delayed and explicit cues, such as future transcripts or the referent's responses. For example, the referent may respond directly to the initial speaker, or other participants might clarify or reiterate the referent in later turns. Traditional offline MMSI model (Lee et al., 2024a) rely on seeing the future social cues directly, making it not an ideal solution for real-time scenarios where such cues are unavailable. Second, online models must rely entirely on immediate and past signals, where subtle social cues–such as pointing gestures and posture, head turns, or body orientation– are crucial for resolving referents. Figure 1 (1) illustrates such a case: the speaker is gesturing toward the referent with the index finger, serving as critical disambiguating contexts for referent resolution. Since social scenes often involve multiple participants engaged in dynamic interactions, it is difficult to detect such subtle yet informative social signals from raw RGB video frames alone without explicit guidance.

To address these challenges, we propose Online-MMSI-VLM, a novel Vision-Language Model (VLM)-based framework that integrates multi-party conversation forecasting and socially-aware visual prompting. For the first challenge, cognitive studies suggest that humans improve online interpretation by anticipating upcoming social interactions (Epperlein et al., 2022; Ma et al., 2024; Hadley & Culling, 2022). This motivates us to leverage multimodal large language models to anticipate future conversations to enrich social context. Our coarse-to-fine strategy first predicts the identity of the upcoming speaker, followed by generating their likely

utterance. For the second challenge, we propose to enhance the representation of past video by using socially-aware visual prompting, which employs off-the-shelf detectors to extract and highlight the positions, postures, and gestures of participants. This facilitates the model to attentively interpret subtle and complex social interactions in the previous video. By jointly forecasting future dialogue and enriching past visual context, our framework can effectively tackle the challenges of Online-MMSI. We evaluate our approach using two VLM models on three referent identification tasks within two social datasets (Lai et al., 2023). The results indicate that Online-MMSI benefits from multi-party conversation forecasting and socially-aware visual prompting, and Online-MMSI-VLM achieves state-of-the-art performance.

In summary, our contributions are threefold: 1) We propose a new task, Online-MMSI, where the AI assistant must interpret multimodal social interactions and provide immediate feedback using only historical dialogues and videos. 2) We propose Online-MMSI-VLM, a multimodal large language model-based framework that integrates multi-party conversation forecasting and socially-aware visual prompting. To the best of our knowledge, it is the first work to use multimodal large language models for advancing MMSI. 3) Extensive experiments, for three tasks on two social datasets, demonstrate the effectiveness of our approach and show Online-MMSI-VLM establishes a strong benchmark for Online-MMSI.

## 2 Related Works

### 2.1 Multimodal Social Interaction Understanding

Multimodal Social Interaction Understanding (MMSI) aims to interpret complex interactions among multiple participants by leveraging both verbal and non-verbal cues, which has attracted increasing attention in the machine learning and social AI communities. For non-verbal social cues, some studies have attempted to perceive social meaning from visual behaviors such as body gestures (Benitez-Garcia et al., 2021; Liu et al., 2021; Chen et al., 2023a; Kapitanov et al., 2024), gaze patterns (Jindal & Manduchi, 2023; Chong et al., 2020; Grauman et al., 2022; Tafasca et al., 2023), and facial expressions (Zhang et al., 2021; Savchenko, 2023; Zhao et al., 2024; Li et al., 2024b). For verbal social cues, prior works have explored social understanding from linguistic signals such as game-theoretic agents (Feng et al., 2025), speaker intent (Malhotra et al., 2022; Chen et al., 2023c), and dialogue sentiment (Feng et al., 2021). For multimodal social cues, several studies integrate verbal and non-verbal modalities to holistically interpret social interactions, such as recognizing emotions (Cheng et al., 2024b; Lian et al., 2025; 2024), conversational dynamics (Raman et al., 2022; Ryan et al., 2023; Hou et al., 2024), and social situations (Hyun et al., 2023; Guo et al., 2025).

Recently, Li et al. (2024a) integrate the perception capability of vision models and the reasoning capability of LLMs for social relation recognition. Gupta et al. (2024) introduce a novel framework to jointly predict the gaze target and social gaze label for all people in the scene. Lee et al. (2024b) conduct a comprehensive survey, identifying three keys for effective social understanding: multimodal social cues, multi-party dynamics, and beliefs. Lai et al. (2023) introduce a multimodal dataset for modeling persuasion behaviors. Cao et al. (2025) introduce a large-scale dataset for multi-person gesture analysis. Lee et al. (2024a) introduce three tasks to model the fine-grained dynamics between multiple people: speaking target identification, pronoun coreference resolution, and mentioned player prediction. To address these tasks, Lee et al. (2024a) introduce a densely-aligned multimodal framework to capture social cues from transcript and video.

Despite these advances, current research typically focuses on the *offline* setting, where the model can have access to entire video frames and dialogues to make predictions. However, this setup does not align well with real-time requirements in interactive AI assistant systems, where the model can only leverage historical information (i.e., the *online* setting). Although methods from recent research (Lee et al., 2024a) can theoretically be adapted to online settings through data rearrangement, they do not adequately account for the challenge of missing future social cues and exploiting historical information in the online setting. In our work, we investigate the online setting for MMSI and propose novel methods to address the challenge.

## 2.2 Multimodal Large Language Models

Multimodal large language models (MLLMs) have demonstrated outstanding performance in vision-language tasks (Liu et al., 2023; Team et al., 2024; Team, 2025; Zhang et al., 2023a; Zhu et al., 2023; Zhang et al., 2023b; Chen et al., 2023b; Lin et al., 2023; Chen et al., 2024b; Cheng et al., 2024c; Zhang et al., 2024b; Thawakar et al., 2025; Zhang et al., 2024c). This success is attributed to their strong perception and reasoning capabilities, which makes them useful for dynamic multimodal social interaction understanding.

**Online scenarios** attract growing attention (Wang et al., 2021; Zhao & Krähenbühl, 2022; Girdhar & Grauman, 2021; Zhao et al., 2023). While several online MLLMs have been developed for training and deployment (Chen et al., 2024a; Liu et al., 2024; Fu et al., 2025), our work aims to conduct Online-MMSI.

**Conversation forecasting** has been explored for conversation agents (Hassan et al., 2024; El Hattami et al., 2023), predicting derailment (Chang & Danescu-Niculescu-Mizil, 2019), and harmful behaviors (Sicilia & Alikhani, 2024). These studies mainly focus on conversation between agents and users. Instead, we focus on understanding multi-party conversations of multimodal social events.

**Visual prompting** has also proven effective in enhancing MLLM performance (Wu et al., 2025; Cai et al., 2024; Wu et al., 2024; Lei et al., 2024; Yang et al., 2023; Xu et al., 2024). Some studies show that attentively leveraging the history is crucial for improving online video understanding in noisy and large-volume visual data (Yao et al., 2025; Zhang et al., 2024a; Huang et al., 2024; Chandrasegaran et al., 2024; Patil et al., 2024). In this work, we propose to leverage visual prompting to enhance historical video for Online-MMSI.

## 3 Problem Formulation and Challenges

Following Lee et al. (2024a), we study three multimodal social interaction understanding tasks: Speaking Target Identification (STI), Pronoun Coreference Resolution (PCR), and Mentioned Player Prediction (MPP), using recorded video and its corresponding multi-party dialogue transcripts. As shown in Fig. 5, STI aims to identify who the current speaker is talking to when the utterance contains a second-person reference, e.g., "you" and "your"; PCR focuses on resolving which participant a third-person pronoun refers to, e.g., "he", "she", "him", "her" and "his"; MPP requires predicting which participant is referred to by name when the participant name in the current utterance is masked, e.g., "So we vote for [MASK]".

The online task formulation is as follows: At timestep $t$, the system receives:

- **User prompt** ($P_t$): a query related to the specific task. The prompts for the three social tasks are "Identify which player the speaker is talking to?", "Determine which player a pronoun refers to?", and "Predict which player is mentioned by name?", respectively.

- **Historical dialogues** ($\mathcal{D}_{(t-d):t}$): the most recent $d$ time steps of dialogue history. Each entry consists of both the speaker identity and the corresponding utterance. E.g., "[Player0]: So, you're a mason? [Player0]: I'm the troublemaker. [Player3]: Did you swap me with anybody?"

- **Recorded video frames** ($\mathcal{V}_{(t-d):t}$): the most recent $d$ time steps of video frames involving multiple participants. These frames include dynamic non-verbal social cues, such as gestures and gazes.

The AI assistant is required to produce an immediate reply:

- **Response** ($R_t$): the predicted referent identity to the referent identification query, which is expressed as a categorical label from a closed set of participants, i.e., "Player0", ..., "PlayerN".

Given $P_t$, $\mathcal{D}_{(t-d):t}$, and $\mathcal{V}_{(t-d):t}$, the objective is to optimize the model to generate an accurate $R_t$. A prediction is considered correct if the predicted identity exactly matches the ground-truth referent.

In the offline setting, the model has access to both past and future data–$P_t$, $\mathcal{D}_{(t-d):(t+d)}$, and $\mathcal{V}_{(t-d):(t+d)}$–and generates a response $R_{(t+d)}$ at timestep $t+d$. In contrast, the online setting restricts access to only historical video and dialogue, cutting off any future context. To quantify the impact of this constraint, Figure 1 (2)

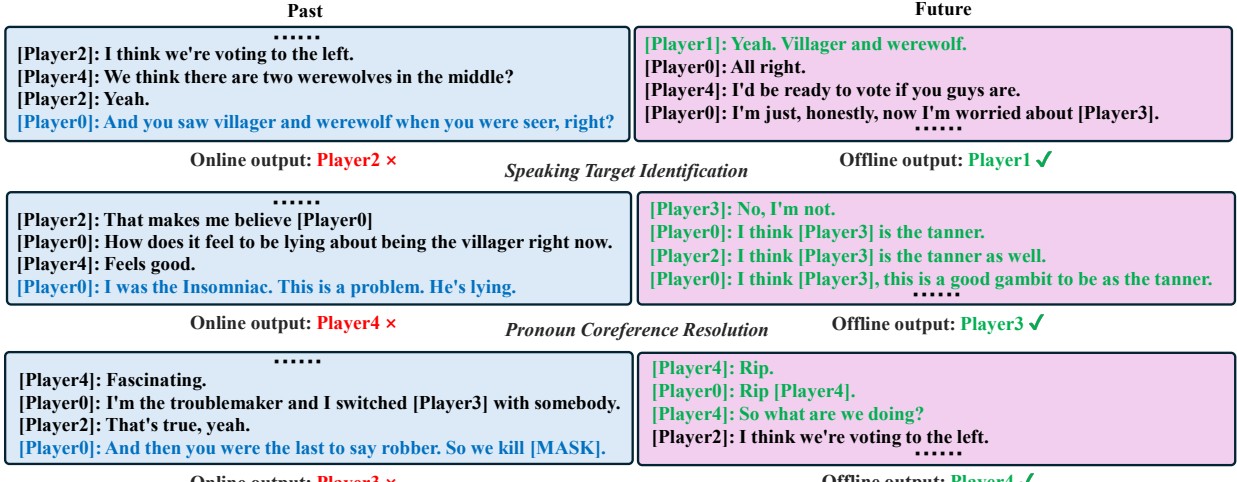

Figure 2: Illustration of the challenge of missing future information when transitioning from the offline setting to the online setting. In the online setting, only immediate (blue font) and past data (black) is available, whereas the offline setting additionally includes useful future context (green), such as the referent's responses. As a result, the online model may be confused and provide an incorrect answer.

compares performance across the three social tasks under both settings using the YouTube dataset (Lai et al., 2023). We evaluate the previous method (Lee et al., 2024a), human participants, GPT-4o (OpenAI, 2024), and Gemini 2.5 Pro (Comanici et al., 2025). To ensure fairness in the user study, each participant first viewed the online input, followed by the offline version. This design mitigates potential bias by preventing participants' online judgments from being influenced by future context seen in the offline setting.

As shown in the results, there is a considerable drop in accuracy across all three tasks when both humans and models when transitioning from offline to online scenarios. The prior model (Lee et al., 2024a) suffers 13.6% in speaking target identification, 2.5% in pronoun coreference resolution, and 11.5% in mentioned player prediction. The average accuracy of humans, GPT-4o, and Gemini 2.5 Pro drops by around 7%, 5%, and 10%, respectively, confirming that the performance gap stems from task difficulty. The evaluation results of GPT-4o and Gemini 2.5 Pro underscore that current LLMs still struggle with understanding multimodal multi-party social interaction compared with humans (Inoue et al., 2025; Tan et al., 2023).

This performance degradation arises from two fundamental limitations. First, online models lack access to delayed and explicit cues–such as future transcripts or the referent's responses–that are available in offline settings. The referent frequently replies to the speaker or is clarified by other participants in subsequent turns. For instance, as seen in Figure 2, in speaking target identification, after the speaker says "[Player0]: And you saw villager and werewolf when you were seer, right?", the referent later replies "[Player1]: Yeah. Villager and werewolf." Offline models can leverage such useful future context to correctly identify "Player0" as the referent. In contrast, online models must rely solely on historical information, which may be ambiguous. For example, in mentioned player prediction, before the speaker says "[Player0]: And then you were the last to say robber. So we kill [MASK].", a previous utterance states "[Player0]: I'm the troublemaker and I switched [Player3] with somebody." The online model may be confused and incorrectly answer "Player3".

Second, online models must resolve social references based entirely on immediate and past signals, where subtle social cues–such as pointing gestures, posture, head turns, and body orientation–become essential. However, in complex interactions involving multiple participants, such signals are often too subtle to interpret from raw RGB video alone, especially without structured visual guidance. Experimental results, as shown in Figure 3, indicate that VLMs fail to capture detailed and accurate social cues from raw RGB video. When prompted to describe the social interaction cues in the video, the VLM typically returns only coarse attributes–such as the relative positions and appearances–using phrases like "on the far right" and "shaved head and a mustache," and occasionally produces hallucinations, such as incorrectly stating "four people."

**Discussion.** Online-MMSI differs fundamentally from history-based egocentric AI tasks (Grauman et al., 2022; Yang et al., 2025; Li et al., 2025; Zhou et al., 2025; Hong et al., 2025; Ye et al., 2024; Cheng et al., 2024a)

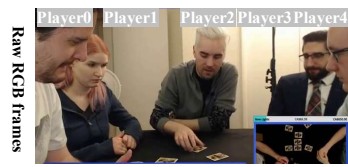

The scene shows **four people**, two men and two women, sitting around a table. **The player on the far left is a woman with red hair. The player next to her is a man with a bald head and a beard. The player next to him is a man with a shaved head and a mustache. The player on the far right is a man with a beard.**

VLM's description the social interaction cues in the video.

Figure 3: Illustration of the challenge in relying solely on immediate and past context when transitioning from offline to online settings, where subtle social cues become increasingly critical. In complex multi-party interactions, such cues are often too nuanced to be reliably inferred from raw RGB video alone, especially in the absence of explicit visual prompting. When asked to describe social interactions, the VLM frequently produces coarse spatial references (green), generic appearance descriptions (yellow), or even hallucinated content (red), while failing to capture the subtle and intricate social signals in multi-person dynamics.

in its modeling objective. Online-MMSI focuses on high-level social interaction understanding, requiring the model to infer who is being referred to in the current utterance by reasoning over verbal and non-verbal social cues. In contrast, retrospective egocentric tasks are typically formulated as event retrieval or grounding problems, answering questions such as when, where, or what happened. Moreover, Online-MMSI emphasizes short-horizon situation modeling, as referential intent is usually resolved through local and rapidly evolving social context, whereas egocentric tasks rely on long-horizon memory retrieval over extended video histories. Finally, Online-MMSI explicitly requires consistent multi-person identity tracking, since the output is a specific participant identity, an assumption that is well-defined in fixed-view multi-party interactions but generally ill-posed in first-person egocentric videos with frequent viewpoint changes.

## 4 Methodology

To address these challenges, we introduce Online-MMSI-VLM, a framework designed to respond to social tasks using only historical input. As illustrated in Figure 4, the model takes as input the user prompt, historical utterances, and video frames, and produces a real-time response under the online setting. To compensate for the lack of future information and enhance the quality of historical signals, we propose two key components: (i) *multi-party conversation forecasting*, which anticipates future dialogue turns; and (ii) *socially-aware visual prompting*, which guides the model to attend to socially relevant visual cues.

### 4.1 Multi-party Conversation Forecasting

To enable forecasting multi-party conversations, we append a forecasting query after the task prompt, which is: "Predict the upcoming speakers' turns and then predict the upcoming utterance of each speaker." After answering the Online-MMSI task, the model generates the next few speaker labels and utterances. Directly predicting the full utterances of each future speaker in a single step can be overly complex and may lead to sub-par generation. Instead, we propose a coarse-to-fine approach with the following two phases:

- **Speaker-turn prediction**: The model first generates a sequence of four upcoming speaker identities, e.g., "The upcoming speakers' turns: Player0, Player4, Player0, Player3."

- **Utterance refinement**: For each predicted speaker, the model then produces a fine-grained utterance. For instance: "The upcoming utterances: [Player0]: I didn't swap you. [Player4]: I was the Insomniac. I did not wake up as myself. So I... [Player0]: Yes. [Player3]: Who did you swap?"

This strategy mimics human conversation patterns, where we first anticipate who might speak and in what order, and then try to guess what they would say. The above process can be formulated as:

$$\hat{S}_{t+1:t+K} = f_\theta^{\text{VLM}}(\mathcal{D}_{(t-d):t}), \tag{1}$$

$$\hat{U}_{t+1:t+K} = f_\theta^{\text{VLM}}(\hat{S}_{t+1:t+K}, \mathcal{D}_{(t-d):t}), \tag{2}$$

where $\hat{S}_{t+1:t+K}$ denotes the predicted sequence of speakers over the next $K$ turns, and $\hat{U}_{t+1:t+K}$ represents the generated sequence of corresponding utterances for each speaker.

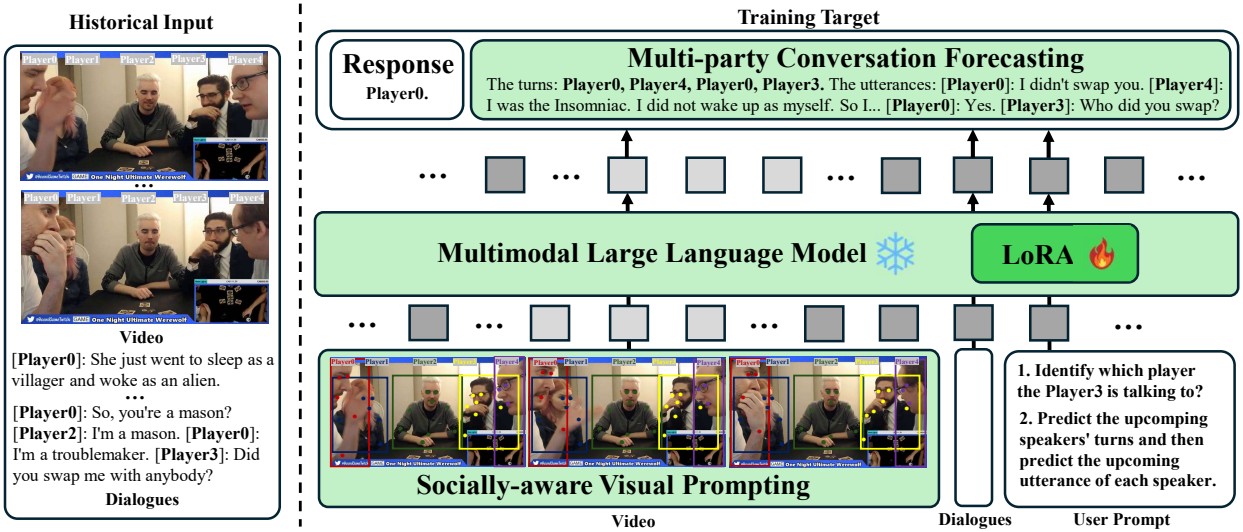

Figure 4: The training pipeline of Online-MMSI-VLM. The model takes the user prompt, historical dialogues, and recorded video as input and generates an immediate response. To tackle the challenges of Online-MMSI, we introduce two techniques: (i) *multi-party conversation forecasting* to enrich language context, and (ii) *socially-aware visual prompting* to facilitate historical social cues modeling.

## 4.2 Socially-aware Visual Prompting

Since online models must rely on immediate and past information, capturing subtle visual cues is crucial for resolving referents. Social scenes often involve multiple participants engaged in dynamic non-verbal interactions, making it difficult to localize and interpret key visual cues such as posture and gestures. Without explicit guidance, the model may fail to attend to the socially salient regions in raw RGB image input, especially in cluttered situations. To address this, we propose socially-aware visual prompting, which explicitly highlights important social cues in the historical video frames. Specifically, we annotate all visible participants with: (a) speaker labels, aligning players with their spoken utterances; (b) bounding boxes, indicating each participant's spatial position; and (c) upper body keypoints, capturing posture, gaze direction, facial expressions, and gestures. These annotations are generated using AlphaPose (Fang et al., 2022).

As shown in Figure 4, we integrate these annotations by overlaying them directly onto the video frames. To distinguish between individuals, we assign a unique color to each person's annotation and include a system prompt specifying the mapping: "The red, blue, green, yellow, purple, and orange colors correspond to Player0, Player1, Player2, Player3, Player4, and Player5, respectively." The annotated frames $\hat{\mathcal{V}}_{(t-d):t}$ are fed into the visual encoder of the VLM, providing a more semantic representation of the social scene. This representation is then concatenated with the tokenized historical dialogues $\mathcal{D}_{(t-d):t}$ and the user prompt $P_t$. The final multimodal embedding is processed by the language head to produce a response $R_t$:

$$R_t = f_\theta^{\text{VLM}}\big([\hat{\mathcal{V}}_{(t-d):t}, \mathcal{D}_{(t-d):t}, P_t]\big), \tag{3}$$

where $R_t$ represents the predicted speaking target, the referent of a pronoun, or the player mentioned by name. By visually highlighting key social signals, this prompting approach helps the model focus on the most relevant regions, improving its ability to interpret past visual context in the absence of future cues.

## 4.3 Supervised Instruction Tuning

To enhance the VLM's performance on specific social tasks, we first construct instruction–output pairs using existing social datasets (Lee et al., 2024a), incorporating future utterances from transcripts into these pairs. We then fine-tune the model via supervised learning with two objectives: (a) task-specific objective: the

model learns to produce correct answers $\hat{y}_t$ for a specific Online-MMSI task; (b) forecasting objective: the model learns to anticipate future dialogue turns $\hat{S}_{t+1:t+K}$ and corresponding utterance content $\hat{U}_{t+1:t+K}$.

Let $\mathcal{L}_{\text{mmsi}}$ and $\mathcal{L}_{\text{forecast}}$ denote the loss for Online-MMSI and forecasting. Our training objective is:

$$\mathcal{L}_{\text{total}} \; = \; \mathcal{L}_{\text{mmsi}} \; + \; \mathcal{L}_{\text{forecast}}. \tag{4}$$

This multi-task formulation enables the model to not only answer current queries but also to anticipate future interactions, leading to better context understanding and improved performance in online social scenarios.

## 5 Experiments

### 5.1 Implementation

We select **LLaMA-3.2-Vision-11B** (Dubey et al., 2024) and **Qwen2.5-VL-7B** (Team, 2025) as our multimodal large language models, both of which are representative state-of-the-art models. For Qwen2.5-VL-7B, we dynamically sample video at a rate of 1 frame per second (fps) as visual input. For LLaMA-3.2-Vision-11B, we sample six frames from each video clip evenly and arrange them into a single $3 \times 2$ grid-like image, following the suggestion from prior work (Kim et al., 2024). We apply instruction tuning with LoRA (Hu et al., 2022), targeting the query and value projection layers, to further specialize these models for our tasks. Specifically, we use an alpha value of 16, a dropout rate of 0.05, and a rank of 512 for the LoRA configuration. We train all new or adapted parameters with a unified cross-entropy loss. The weights of two losses are set as 1 by default. The learning rate is set to $1 \times 10^{-4}$ empirically for the speaker target identification and mentioned player prediction tasks, and $1 \times 10^{-3}$ for the pronoun coreference resolution task. The batch size is set to 1, with gradient accumulation steps of 4. Each model is trained for 5 epochs on all three tasks. The historical window length $d$ is set to 10 dialogue turns, and the forecasting horizon $K$ is set to 4 utterance turns. The average duration of each dialogue turn is approximately 2.8 seconds. The tracking information and keypoints for prompting are generated by AlphaPose (Fang et al., 2022) and curated using the reference frame from Lee et al. (2024a). More experiment implementations are provided in Appendix section.

### 5.2 Datasets

Experiments are conducted on the Werewolf Among Us dataset, which comprises two subsets (YouTube and Ego4D) of social deduction games (Lai et al., 2023). We follow the dataset and task setup of Lee et al. (2024a), while cutting the future video and transcript in each sample for the online setting.

**YouTube** contains 151 games of One Night Ultimate Werewolf, which corresponds to 151 separate videos with 14.8 hours and transcripts comprising 20,832 utterances. It has 3,255 samples for speaking target identification, 2,679 for pronoun coreference resolution, and 3,360 for mentioned player prediction.

**Ego4D** has 40 games of One Night Ultimate Werewolf and 8 games of The Resistance: Avalon. It contains 101 separate videos with 7.3 hours and transcripts containing 5,815 utterances, with 832 samples for speaking target identification, 503 for pronoun coreference resolution, and 472 for mentioned player prediction.

**Evaluation.** We follow Lee et al. (2024a) to report the overall accuracy of the predicted referent.

### 5.3 Performance Comparison

Table 1 presents the performance of our proposed Online-MMSI-VLM framework with two VLM backbones–Qwen2.5-VL and LLaMA-3.2-V–compared to a wide range of baselines across three social understanding tasks: speaking target identification, pronoun coreference resolution, and mentioned player prediction. Evaluations are conducted on both the YouTube and Ego4D subsets. We compare against capable VLMs such as InternVL3 (Zhu et al., 2025), Qwen2.5-VL and LLaMA-3.2-V under vanilla finetuning. We also include results from Lee et al. Lee et al. (2024a) trained offline / online and tested online. Both variants of Online-MMSI-VLM consistently outperform the baselines across all tasks and datasets. Notably, the LLaMA-based model achieves the highest average accuracy on YouTube (60.6%) and Ego4D (56.1%). We report the zero-shot performance of Gemini 2.5 Pro and GPT-4o. Interestingly, their average zero-shot accuracy on Ego4D

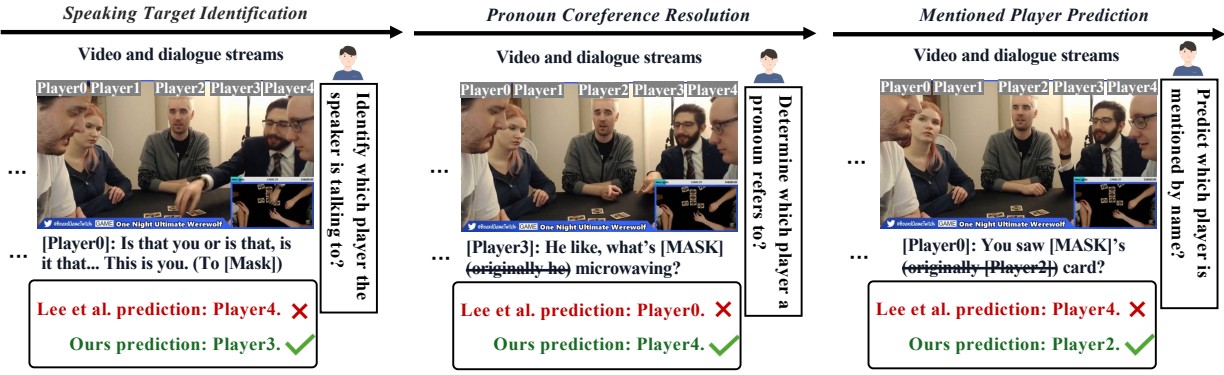

Figure 5: Qualitative results of Online-MMSI-VLM on three social tasks. Compared to the prior work (Lee et al., 2024a), our method provides immediate and accurate referents based on historical video and dialogue.

Table 1: Performance comparison of baselines and Online-MMSI-VLM on YouTube / Ego4D across three MMSI tasks: Speaking Target Identification (STI), Pronoun Coreference Resolution (PCR), and Mentioned Player Prediction (MPP). Results are reported as Accuracy (%) on YouTube / Ego4D.

| Model | STI | PCR | MPP | Average Accuracy |
|---|---|---|---|---|
| Lee et al. (2024a) (Offline) | 60.0 / 52.8 | 63.5 / 41.1 | 43.2 / 32.1 | 55.6 / 42.0 |
| Lee et al. (2024a) (Online) | 59.1 / 59.6 | 63.4 / 55.4 | 47.3 / 41.0 | 56.6 / 52.0 |
| Vanilla InternVL | 62.4 / 52.8 | 65.1 / 55.2 | 47.0 / 36.8 | 58.2 / 48.2 |
| Vanilla Qwen | 60.8 / 57.7 | 62.9 / 48.2 | 45.8 / 32.9 | 56.5 / 46.3 |
| Vanilla LLaMA | 63.2 / 60.2 | 65.4 / 54.5 | 46.6 / 38.2 | 58.4 / 51.0 |
| Gemini 2.5 Pro (Zero-shot) | 23.3 / 30.5 | 26.7 / 29.4 | 16.7 / 30.2 | 22.3 / 30.0 |
| GPT-4o (Zero-shot) | 38.5 / 44.0 | 35.4 / 50.9 | 37.9 / 31.6 | 37.3 / 42.2 |
| Online-MMSI-VLM (Qwen) | 64.6 / 61.7 | **68.8** / **60.7** | 47.2 / 39.5 | 60.2 / 53.9 |
| Online-MMSI-VLM (LLaMA) | **64.9** / **65.1** | 68.5 / 59.8 | **48.4** / **43.4** | **60.6** / **56.1** |

is higher than on YouTube, while the trained models show the opposite trend. This can be explained by the fact that YouTube contains substantially more training samples than Ego4D, which boosts the performance of fine-tuned models but does not affect zero-shot results. The streaming video understanding models, such as Chen et al. (2024a) and Liu et al. (2024), are not directly suitable for comparison due to fundamental differences in objectives. Our work focuses on Online-MMSI, which requires reasoning over multi-party social dynamics and explicitly handling the absence of future context. But streaming video understanding models are mainly designed for efficient processing in general-purpose video tasks and are built upon similar backbones as ours (i.e. a LLaMA-3-8B language model). They don't have mechanism to explicitly capture the social interaction cues (e.g., gestures and postures) nor deal with the challenge of missing future context.

Across tasks, we observe that pronoun coreference resolution generally achieves the highest accuracies (up to 68.8%), followed by speaking target identification, while mentioned player prediction remains the most challenging, with accuracies typically below 50%. This pattern can be explained by the differences how each task depends on context. Both pronoun coreference and speaking target identification often involve referents located in adjacent utterances–either the preceding or following turn–making them more amenable to forecasting and short-range historical modeling. In contrast, we find mentioned player prediction typically requires tracking entities introduced several turns earlier, as the referent is often not mentioned in the immediate context. This temporal gap increases the difficulty of reasoning mentioned player.

Figure 5 illustrates qualitative comparison examples of the three social tasks in streaming video, demonstrating how our method leverages only past dialogue turns and annotated frames to accurately identify social references. For user queries such as "Predict which player is mentioned by name," "Determine which player a pronoun refers to," and "Identify which player the speaker is talking to," our model consistently provides immediate and accurate referents based solely on prior context. In contrast, the prior method (Lee et al., 2024a) often struggles in online settings due to a lack of tailored design for these challenges.

Table 2: Ablation study of proposed components for three social tasks on the YouTube and Ego4D datasets. Both socially-aware visual prompting and multi-party conversation forecasting consistently improve performance over the baseline. Their combination yields the best results across all three tasks and both datasets.

| Model | Forecasting | Prompting | Speaking Target Identification | | Pronoun Coreference Resolution | | Mentioned Player Prediction | |
|---|---|---|---|---|---|---|---|---|
| | | | YouTube | Ego4D | YouTube | Ego4D | YouTube | Ego4D |
| Qwen | - | - | 60.76 | 57.71 | 62.88 | 48.21 | 45.83 | 32.89 |
| Qwen | - | ✓ | 61.45 | 58.10 | 64.43 | 58.93 | 46.24 | 35.52 |
| Qwen | ✓ | - | 63.96 | 60.00 | 65.20 | 54.46 | 46.52 | 36.84 |
| Qwen | ✓ | ✓ | **64.58** | **61.71** | **68.83** | **60.71** | **47.20** | **39.47** |
| LLaMA | - | - | 63.20 | 60.20 | 65.39 | 54.46 | 46.61 | 38.15 |
| LLaMA | - | ✓ | 64.02 | 64.00 | 67.87 | 55.35 | 47.33 | 42.10 |
| LLaMA | ✓ | - | 64.43 | 61.71 | 67.30 | 55.35 | 47.34 | 40.78 |
| LLaMA | ✓ | ✓ | **64.88** | **65.14** | **68.45** | **59.82** | **48.43** | **43.42** |

Table 3: Performance impact of forecasting components on three social tasks using Qwen-based model on the YouTube dataset. Detailed future utterance generation consistently enhances results, while adding speaker-turn prediction yields further improvement, demonstrating the strength of a coarse-to-fine strategy.

| Speaker Turns | Detailed Utterances | Speaking Target Identification | Pronoun Coreference Resolution | Mentioned Player Prediction |
|---|---|---|---|---|
| - | - | 60.76 | 62.88 | 45.83 |
| ✓ | - | 60.31 | 64.63 | 46.39 |
| - | ✓ | 62.69 | 64.92 | 45.93 |
| ✓ | ✓ | **63.96** | **65.20** | **46.52** |

## 5.4 Effects of Conversation Forecasting

Table 2 shows the effects of multi-party conversation forecasting, where forecasting consistently improves performance across all tasks. For example, in the pronoun coreference resolution task on the YouTube dataset, applying forecasting boosts Qwen-based's accuracy from 62.88% to 65.20%. Similarly, for speaking target identification, forecasting improves Qwen-based's performance from 60.76% to 63.96%. These results confirm that anticipating future conversational turns and utterances helps the model compensate for the lack of future context in the online setting. The improvement for mentioned player prediction is relatively smaller, possibly because this task, mentioning someone in a dialogue, relies less on future context.

Table 3 further analyzes the impact of the two components in our coarse-to-fine forecasting framework: speaker-turn prediction and detailed utterance generation. Results indicate that enabling both components yields the highest overall performance. In particular, direct detailed utterance generation consistently enhances accuracy by enriching the context. Moreover, integrating speaker-turn prediction provides additional gains, suggesting that predicting turn structure first facilitates the generation of relevant utterances.

Table 4 explores the effects of different forecasting turn lengths (i.e. 2, 4, or 8 future turns). Empirically, a 4-turn forecasting length provides the most reliable performance gains across the three social tasks on both datasets. From the observation in 5.6, multi-party conversation forecasting is challenging due to its uncertainty nature. Extending the forecasting length further increases the risk of hallucinated, question-irrelevant content and adds computational complexity, while contributing little additional useful context. Thus, longer forecasts do not improve performance linearly; instead, they exhibit diminishing returns. For these reasons, forecasting 4 turns strikes the most practical balance between accuracy and complexity.

## 5.5 Effects of Visual Prompting

Table 2 also shows the effects of socially-aware visual prompting, where prompting further enhances model performance by effectively enhancing past information. For example, in the pronoun coreference resolution task, adding visual prompting raises Qwen-based's accuracy from 62.88% to 64.43% and LLaMA-based's

Table 4: Impact of forecasting length on three social tasks using Qwen-based model across YouTube and Ego4D datasets. Forecasting 4 future turns typically yields the highest performance gains across all tasks.

| Dataset | YouTube (%) | | | Ego4D (%) | | |
|---|---|---|---|---|---|---|
| Forecast Length (Turns) | 2 | 4 | 8 | 2 | 4 | 8 |
| Speaking Target Identification | 62.44 | **64.58** | 61.98 | **62.03** | 61.71 | 59.43 |
| Pronoun Coreference Resolution | 68.26 | **68.83** | 67.11 | 56.07 | **60.71** | 55.54 |
| Mentioned Player Prediction | 45.98 | **47.20** | 46.25 | 34.26 | **39.47** | 39.47 |

Table 5: Effects of different visual prompting modules on three social tasks. We adopt the Qwen-based model and evaluate on the YouTube dataset. Incorporating bounding boxes (BB) or upper-body keypoints (KP) improves average accuracy over raw RGB frames, and combining them yields the best results.

| Visual Cues | Speaking Target Identification | Pronoun Coreference Resolution | Mentioned Player Prediction | Avg. Accuracy |
|---|---|---|---|---|
| Raw RGB frames | 63.96 | 65.20 | 46.52 | 58.56 |
| RGB + BB | 64.13 | 68.13 | 46.71 | 59.66 |
| RGB + KP | 63.96 | 65.79 | 47.10 | 59.28 |
| RGB + BB + KP | **64.58** | **68.83** | **47.20** | **60.20** |

from 65.39% to 67.87% on YouTube. This result verifies that visual prompting can serve as highlighting the crucial areas in the video, enabling VLMs to capture subtle and informative social cues in historical video.

We investigate the impact of incorporating different types of visual annotations–namely, bounding boxes and upper body keypoints–on model performance. In the baseline setting, raw RGB frames are annotated only with speaker identities to align visual input with the corresponding utterances. As shown in Table 5, incorporating bounding boxes and keypoints improves average accuracy over raw RGB frames by 1.1% and 0.72% respectively, and further combining bounding boxes and keypoints yields the highest gain of 1.64%. It indicates that bounding boxes and keypoints stress complementary social cues at different levels of granularity. Bounding boxes primarily highlight coarse-level information, such as participants' spatial location and body orientation. In contrast, upper-body keypoints emphasize finer-grained signals, including gaze direction and hand gestures. These improvements demonstrate that explicitly highlighting players' spatial positions and gestures enables the model to better interpret social dynamics within the scene.

The tracking and keypoints are provided by Lee et al. (2024a), where annotations are generated by Alpha-Pose (Fang et al., 2022) followed by human curation with the reference frame. To further investigate model robustness to inaccurate tracking and keypoints, we conducted a preliminary study for visual prompting. During inference, we explored 20% dropout to keypoints or 10% dropout to tracking, simulating keypoint jitter (motion/occlusion) and short-term tracking loss. Our Qwen-based approach maintains strong performance on the speaker target identification task, with 64.5% on Youtube and 61.7% on Ego4D.

Furthermore, as shown in Table 2, combining socially-aware visual prompting and multi-party conversation forecasting consistently yields the best performance across all three tasks and both datasets. This demonstrates the complementary strengths of forecasting future conversations and enriching visual understanding.

### 5.6 Evaluation of Generated Conversations

Figure 6 presents two examples of generated future speaker turns and utterances, where green text indicates accurate and consistent conversations with the ground truth and yellow denotes different but reasonable utterances. Although the predicted turns and wording may differ slightly from the ground truth, the model successfully produces plausible speakers and content that align with the surrounding context and human social reasoning. This helps the model build a more coherent understanding of the ongoing dialogue while missing future context. In the first example, the generated utterances reinforce the exchange between Player1

| Previous | Previous |
|---|---|
| [**Player5**]: For real. | [**Player2**]: He could be a drunk. |
| [**Player4**]: Ah, that's a convenient card. | [**Player1**]: Which ones did you look at again? I can't remember. |
| [**Player1**]: I was the Insomniac. | [**Player0**]: I looked at just his card. |
| [**Player5**]: What were you now? | [**Player1**]: Just his card. |

| Ground Truth | Ground Truth |
|---|---|
| [**Player1**]: The Insomniac. | [**Player0**]: Yes. |
| [**Player5**]: That's funny because I'm the Insomniac. | [**Player1**]: And it said it was a werewolf? |
| [**Player1**]: That's funny because I'm the Insomniac. | [**Player2**]: I suddenly don't think you're a seer. |
| [**Player4**]: They have the same story. | [**Player1**]: His said werewolf? |

| Forecast | Forecast |
|---|---|
| [**Player1**]: I was the Insomniac. | [**Player0**]: Yeah. |
| [**Player5**]: So you're saying he's lying? | [**Player1**]: Okay. What did you see? |
| [**Player1**]: Yeah. | [**Player0**]: I saw a werewolf. |
| [**Player5**]: And you're saying you're the Insomniac? | [**Player2**]: That's a little suspicious. |

Figure 6: Samples of ground truth future conversations and generated forecasts. Although the predicted turns and wording may differ slightly from the ground truth (yellow), the model successfully produces plausible speakers and content that align with the surrounding context and human social reasoning (green).

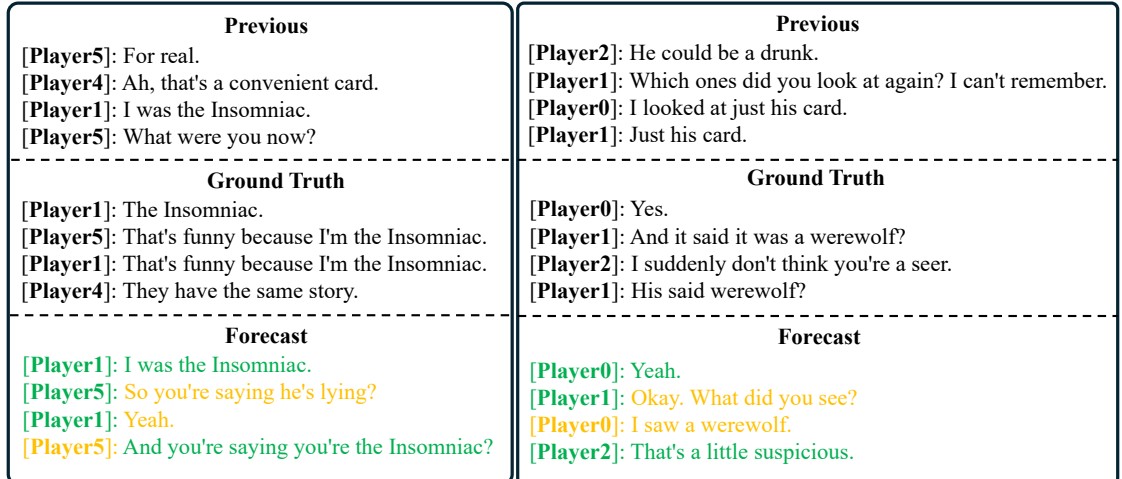

Figure 7: Failure case of the forecasting module on the STI task. The predictions align with historical context but deviate from the ground truth, illustrating the challenge of anticipating uncertain social dynamics. Notably, the goal of forecasting is to enhance social understanding, rather than exact future prediction.

and Player5 regarding the Insomniac role. In the second, the forecasting captures a multi-party interaction among Player0, Player1, and Player2 concerning the observation of another player's card.

To further measure the quality of the generated conversations, we employ Macro F1 (Pedregosa et al., 2011) and BERTScore (Zhang et al., 2019) as evaluation metrics. Macro F1 measures the accuracy of predicted speaker turns, while BERTScore evaluates the semantic similarity between generated and ground-truth utterances. We exclude BLEU (Papineni et al., 2002) from our evaluation due to its reliance on exact n-gram matches, which makes it less reliable for assessing short and diverse conversational responses, where minor paraphrasing or changes in word order can disproportionately penalize the score. Our approach achieves a Macro F1 of 0.68 and a BERTScore of 0.86, indicating accurate upcoming speaker prediction and strong semantic alignment with the ground truth. These results validate the effectiveness of our proposed conversation forecasting in enhancing linguistic context for Online MMSI.

Figure 7 shows a failure case of forecasting where forecasts are consistent with history but diverge from ground truth, highlighting the challenge of anticipating uncertain social dynamics. Through these experiments and evaluation, we observe that multi-party conversation forecasting remains a challenging task due to the inherent unpredictability of real-world dialogue and the current limitations of LLMs in complex social reasoning. Nevertheless, our goal is not to achieve precise forecasting, but to use the process as a means of enhancing social understanding. Interestingly, we observe that even inaccurate forecasts can improve performance–possibly because forecasting encourages the model to role-play the historical dialogue, thereby deepening its understanding of social context and interaction dynamics.

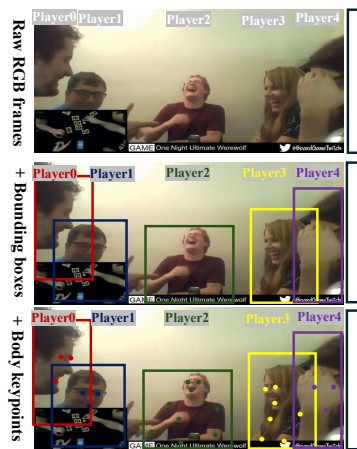

Figure 8: VLM-generated descriptions under different visual prompting settings. The results show that using raw RGB frames yields generic descriptions with limited reference to social interactions. In contrast, when participants are annotated with bounding boxes and body keypoints, the VLM generates more accurate and detailed accounts of social dynamics, such as "nodding and smiling" and "looking at the speaker".

### 5.7 Further Analysis of Visual Prompting

We explored how the model understands the social interaction in the historical video under different visual prompting. We provide Qwen2.5 VL with videos containing different visual promptings and the query: "Describe the social interaction cues in the video." The generated descriptions are presented in Figure 8, where green text indicates accurate social descriptions, red highlights misleading or incorrect descriptions, and black denotes general, non-specific descriptions. The results demonstrate that, by annotating each participant with bounding boxes and body keypoints, the VLM produces more accurate and detailed descriptions of social interactions. For example, the model generates the description: "The person in the yellow box is also smiling and looking at the speaker." Compared to other prompting's generation, such an description has more details about body movement and facial expression. This suggests that socially-aware visual prompting effectively emphasizes critical regions of interest, thereby improving the model's ability to interpret subtle social dynamics in noisy and large historical video. We also noticed that, in raw RGB frames input, the description not only fails to capture detailed social interaction cues but also contains hallucinations: "The person second from the left is holding a drink, while the person second from the right is holding a phone." The visual prompting shows capability of suppressing inaccurate description.

### 5.8 Efficiency Evaluation

To assess Online-MMSI-VLM's real-time applicability, we evaluate the model's latency and throughput. Experiments show the model achieves a median latency of 302 ms and a throughput of 1.99 samples per second when running on a single A6000 GPU, supporting deployment in natural conversational scenarios.

For completeness, we additionally conducted efficiency evaluations in terms of inference latency and GPU memory consumption on representative baselines. Table 6 reports the inference latency of different methods. We observe that Online-MMSI-VLM consistently maintains sub-second latency across both backbones (Qwen and LLaMA), with inference times of 0.302 s and 0.554 s, respectively. This efficiency enables immediate responses in online scenarios. In contrast, API-based methods such as GPT-4o and Gemini 2.5 Pro exhibit higher response latency, which limits their suitability for real-time deployment. Moreover, the substantially higher latency of Lee et al. (2024a) primarily stems from its offline processing pipeline, which relies on future conversational context and delayed processing, making it inherently unsuitable for online inference.

We additionally evaluate GPU memory consumption during inference for open-source vision-language models, as summarized in Table 7. Online-MMSI-VLM requires approximately 20 GB of GPU memory, which is comparable to its vanilla counterparts. Importantly, this memory footprint is well within the capacity of widely adopted embedded computing platforms such as Jetson AGX Orin, which provides 32 GB unified

Table 6: Inference latency comparison across different models. Online-MMSI-VLM maintains sub-second latency, supporting immediate feedback in online settings.

| Model | Latency (s) |
|---|---|
| Lee et al. (2024a) | 14.010 |
| GPT-4o | 3.846 |
| Gemini 2.5 Pro | 2.207 |
| Vanilla Intern VL | 2.953 |
| Vanilla Qwen | 0.268 |
| Vanilla Llama | 0.533 |
| Online-MMSI-VLM (Qwen) | 0.302 |
| Online-MMSI-VLM (Llama) | 0.554 |

Table 7: GPU memory usage during inference for open-source vision-language models. Online-MMSI-VLM requires approximately 20 GB GPU memory, enabling stable real-time inference on Jetson platforms.

| Model | GPU Memory (GB) |
|---|---|
| Vanilla Intern VL | 21.31 |
| Vanilla Qwen | 17.26 |
| Vanilla Llama | 21.49 |
| Online-MMSI-VLM (Qwen) | 17.39 |
| Online-MMSI-VLM (Llama) | 21.52 |

memory. This leaves sufficient headroom for runtime overhead and system-level processes, enabling stable and real-time online inference in practical deployment settings, including robotics and embodied AI applications.

We also evaluated the end-to-end latency of whole system, processing speech and generating forecasts and responses, and the result is 343 ms, demonstrating feasibility for real-time online use. Specifically, in the social game scenario, participants sit together, each equipped with a headset microphone. The streaming audio from each speaker is transcribed by Whisper (Radford et al., 2023), which processes audio at a speed of 41 ms per second. Because transcription is incremental, this latency remains nearly constant regardless of utterance length. Combined with forecasting and generating answer, the end-to-end latency is 343 ms.

### 5.9 Common Failure Cases

Fig. 9 illustrates representative failure cases across the three social interaction tasks. We observe that the model tends to fail in scenarios involving overlapping social dynamics. For example, in the first STI case, two participants (Player2 and Player3) simultaneously address different interlocutors, creating ambiguity in addressee resolution. Similarly, in the second PCR case, the target utterance contains multiple potential referents (e.g., she and him), increasing coreference uncertainty. In addition, the model struggles when social interaction cues are weak or indirect. For instance, in the third MPP case, Player3 talks to Player4 and refers to Player0 with the statement "Player0 is werewolf", while the supporting verbal context ("Player0: I think we kill [Player3].") provides only a weak and indirect cue. In such cases, the lack of strong verbal or non-verbal signals makes correct referent identification particularly challenging.

## 6 Conclusion

We introduce a new task, Online Multimodal Social Interaction Understanding, which requires models to interpret social interactions using only historical information. This setting better aligns with real-world human-AI interaction scenarios, where the AI assistants are required to reply immediately. Since future context is unavailable, the performance of existing models, human, and advanced AI tools degrade significantly. To address this challenge, we propose **Online-MMSI-VLM**, a novel framework built upon recent advanced

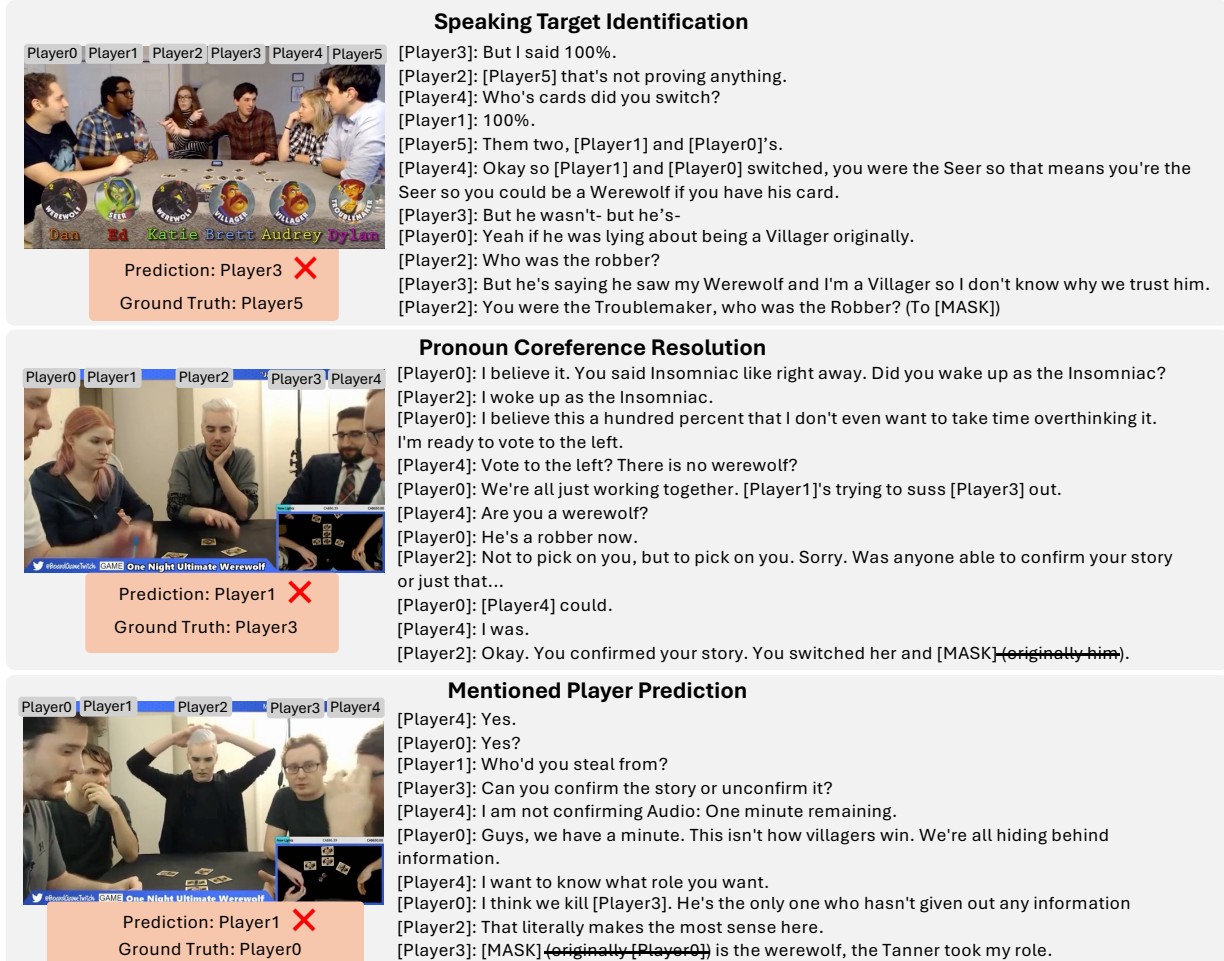

Figure 9: Common failure cases across three social tasks. We observe that the model is likely to fail when social dynamics overlap or when social interaction cues are weak or ambiguous.

MLLMs, integrating techniques: (1) *multi-party conversation forecasting*, which anticipates future dialogue turns to enrich linguistic context, and (2) *socially-aware visual prompting*, which highlights critical visual cues for more accurate social reasoning. Extensive experiments across multiple benchmarks demonstrate the effectiveness of our method in online social understanding tasks. Further ablation studies validate the individual contributions of each proposed component. We believe this work represents an important step toward developing practical and robust social AI systems capable of functioning in realistic environments. Nevertheless, such systems could also be repurposed for surveillance of social interactions, raising significant privacy and ethical concerns. These considerations underscore the need for responsible deployment.

## Acknowledgements

We thank Teng Wang for early-stage inspiration that shaped this line of work. We also thank our colleagues and peers for their valuable feedback and suggestions on this paper.

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

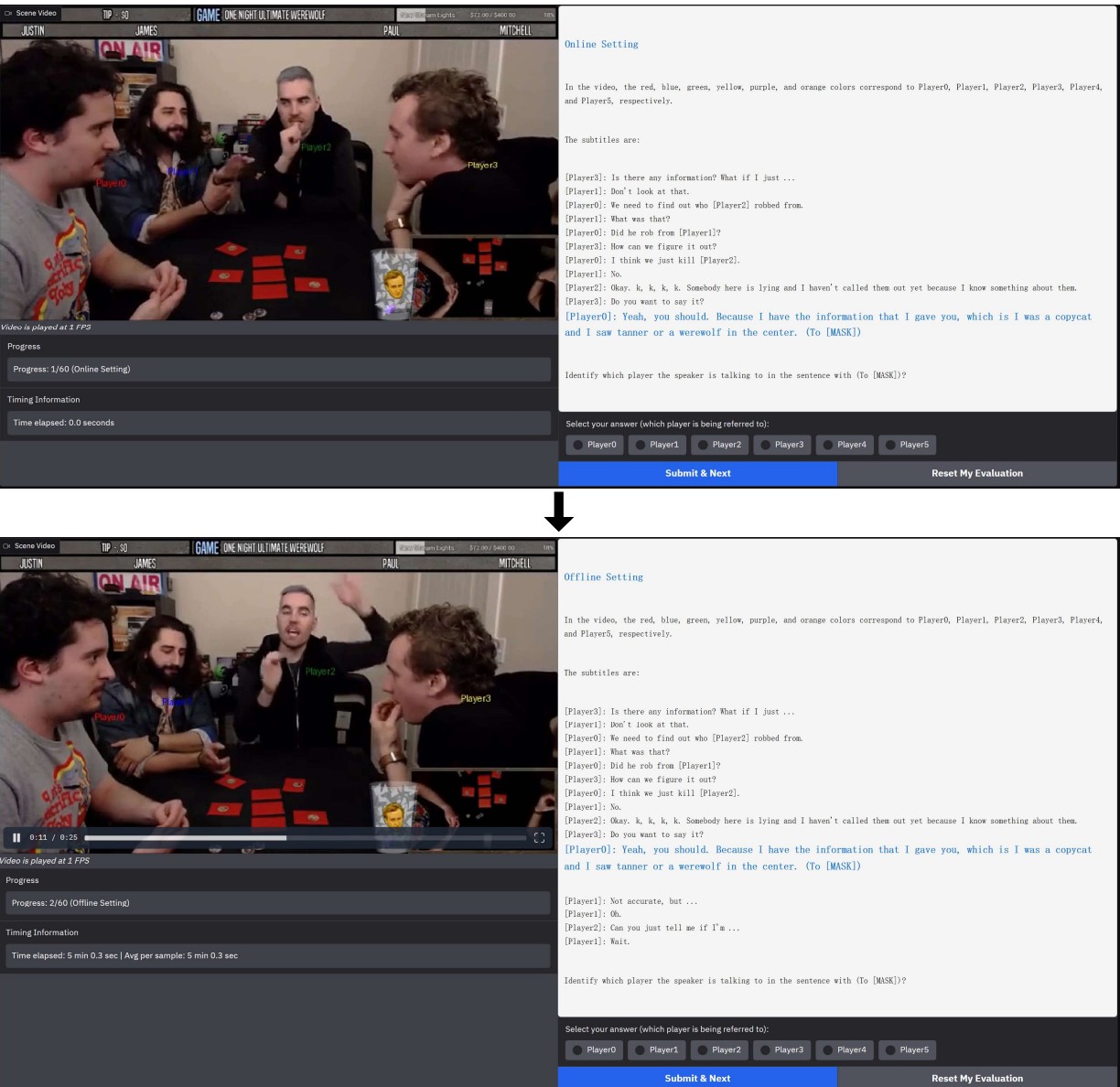

Figure 10: Illustration of human study on online setting and then offline.

# A  Appendix

## A.1  Implementation Details

The training of our video-based model is built upon the Qwen2.5-VL-7B-Instruct[1] (Team, 2025) or LLaMA-3.2-11B-Vision[2] (Dubey et al., 2024) architecture with image splitting disabled. The model operates with bfloat16 precision and employs FlashAttention-2 (Dao et al., 2022) for efficient memory usage. For Qwen2.5-VL-7B-Instruct, the videos are sampled at 1.0 FPS and a resolution constraint of $36 \times 42 \times 10$ pixels per frame; for LLaM-3.2-11B-Vision, we transfer the video into a $3 \times 2$ grid-like image and set the maximum image size as $1120 \times 1120$ pixels. The max new tokens is set as 1024 in both models. A LoRA-based (Hu et al., 2022) fine-tuning strategy is implemented, with LoRA alpha set to 16, a dropout rate of 0.05, a rank of 512, and modifications applied to the query and value projection layers. The training configuration includes

---

[1] https://huggingface.co/Qwen/Qwen2.5-VL-7B-Instruct
[2] https://huggingface.co/meta-llama/Llama-3.2-11B-Vision

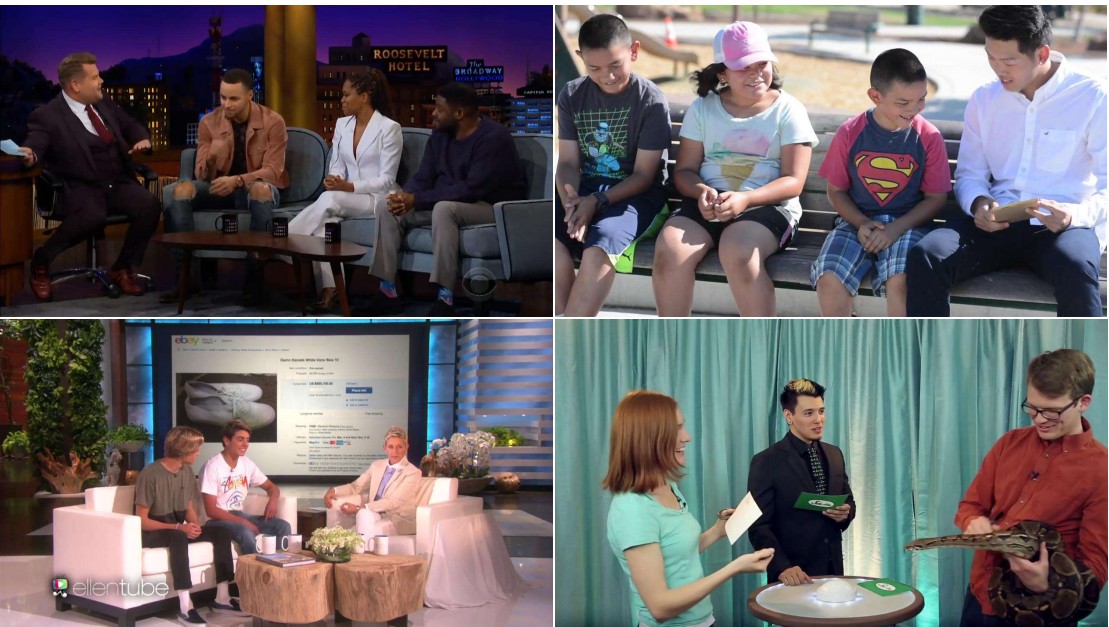

Figure 11: Examples of curated clips from the new evaluation dataset, SIQ. The clips are primarily drawn from talk shows with multiple participants, representing a domain that differs from social deduction games.

a batch size of 1 per device, gradient accumulation steps of 4, and a total of 5 epochs. The optimizer used is AdamW with a fused implementation, and the learning rate follows a linear decreasing scheduler strategy. The regularization weight decay of 0.01 and a gradient clipping threshold of 0.3. Please note that we adopt different learning rates for pronoun coreference resolution because tasks differ in the number of available training samples and annotation density (3,255 for speaker target identification, 3,360 for mentioned player prediction, and 2,679 for pronoun coreference resolution). In practice, we empirically observed that assigning a learning rate of $1 \times 10^{-3}$ to pronoun coreference resolution leads to more stable optimization and improved performance across tasks compared to using a uniform learning rate of $1 \times 10^{-4}$.

## A.2   Human Study

We select 30 samples randomly for each participant and show the online input followed by its offline version, as shown in Figure 10. In this way, the participant gets rid of being affected by the prior offline information.

## A.3   Limitations

While our method achieves strong performance on practical Online-MMSI, several limitations remain. First, the approach relies on external preprocessing steps as previous approaches (Lee et al., 2024a), such as visual tracking and speech transcription, which may introduce errors and affect overall performance. Second, the capabilities of the model are bounded by the social reasoning abilities of current LLMs, which may struggle with complex or subtle social dynamics. Future work may explore end-to-end solutions and more advanced perception and reasoning mechanisms to further enhance online social interaction understanding.

## A.4   Auxiliary Signals

To explore more auxiliary signals for visual prompting, we also extracted gaze signal from videos. However, integrating gaze through visual prompting results in performance drops, i.e., 0.8%, 0.3%, and 0.5% on speaker target identification, pronounce coreference recognition, and mentioned player prediction, respectively. It might be due to inaccurate gaze estimations which are derived from GazeFollowing (Lian et al., 2018). Besides, some studies show existing LLMs might struggle to incorporate gaze information within the context of multi-party dialogues (Inoue et al., 2025), indicating more sophisticated methods are needed.

Table 8: Cross-dataset transfer on speaker target identification (accuracy / gain). These results indicate that the model fine-tuned on social deduction games exhibits robust generalization ability.

| Train → Test | YouTube | Ego4D | SIQ |
|---|---|---|---|
| YouTube | 64.58 (+43.82) | 63.71 (+44.85) | 68.80 (+26.8) |
| Ego4D | 47.18 (+26.42) | 61.71 (+42.85) | 47.20 (+5.2) |

Table 9: Zero-test results on speaker target identification. We can see SIQ appears less challenging than the social game datasets. This is consistent with the domain characteristics: while talk shows involve meaningful host–guest exchanges, the interactions are typically centralized and sequential. In contrast, social games distribute turn-taking among multiple participants, leading to more frequent, multi-directional interactions.

| Test Dataset | YouTube | Ego4D | SIQ |
|---|---|---|---|
| Accuracy | 20.76 | 18.86 | 42.00 |

## A.5 Cross Dataset and Domain Generalization

To assess the generalization capability of Online-MMSI-VLM, we conducted both cross-dataset and cross-domain evaluations. Specifically, we trained the Qwen-based model on either YouTube or Ego4D and tested it on YouTube, Ego4D, and SIQ, respectively. Currently, YouTube and Ego4D subsets are only publicly available datasets (Lee et al., 2024a) designed for MMSI tasks, we curated a new evaluation dataset, SIQ, from the talk show domain, which provides a setting distinct from social deduction games.

The Table 8 reports the accuracy and relative performance gain for speaker target identification under Online-MMSI-VLM (Qwen-based). Here, the reported gain is measured against the zero-shot baseline. As shown, when fine-tuned on YouTube, the model achieves substantial improvements (over 25%) when transferred to Ego4D and SIQ, demonstrating strong generalization across datasets and domains. When fine-tuned on Ego4D, the model's improvement suffers degradation when transferred to YouTube and SIQ. This degradation can be attributed to the limited size of Ego4D (832 samples) compared to YouTube (3,255 samples), which increases the risk of overfitting and weakens generalization. Overall, these results indicate that the model fine-tuned on social deduction games exhibits robust generalization ability.

We provide additional details on the curated SIQ dataset. We selected videos featuring multiparty social interactions from Social-IQ 1.0 (Zadeh et al., 2019). Although Social-IQ 1.0 contains 1,115 videos, most are TV shows with only two participants or lack genuine social interaction. After careful screening, we constructed 125 samples for the speaker target identification task, following the procedure of Lee et al. (2024a). The curated subset has a total duration of 53 minutes and contains 953 utterances. The selected clips, as shown in Figure 11, are primarily drawn from talk shows with multiple participants, representing a domain that differs from social deduction games. The Table 9 shows the zero-shot results, indicating SIQ appears less challenging than the social game datasets. This is consistent with the domain characteristics: while talk shows involve meaningful host–guest exchanges, the interactions are typically centralized and sequential. In contrast, social games distribute turn-taking among multiple participants, leading to more frequent, multi-directional interactions. All videos are publicly available at https://www.kaggle.com/datasets/mathurinache/social-iq.

## A.6 Weights of $\mathcal{L}_{\mathrm{mmsi}}$ and $\mathcal{L}_{\mathrm{forecast}}$

The weights of $\mathcal{L}_{\mathrm{mmsi}}$ and $\mathcal{L}_{\mathrm{forecast}}$ are set to 1 by default, since tokens for MMSI and forecast are treated equally in loss computation. To study the effect of varying their ratio, we fixed the weight of $\mathcal{L}_{\mathrm{mmsi}}$ at 1 and tuned the weight of $\mathcal{L}_{\mathrm{forecast}}$ to 0.2, 0.5, 1, 1.2, 1.5, and 1.8, respectively. Experiments were conducted on the YouTube subset for speaker target identification using Online-MMSI-VLM (Qwen). As shown in Table 10, weight ratios of 1, 1.2, and 1.5 achieve the best results, while both too low and too high ratios lead to degraded accuracy. These results indicate that our default setting of 1 is near-optimal and robust.

Table 10: Impact of loss weight ratio on speaker target identification (YouTube subset). These results indicate that our default setting of 1 for two loss weights is near-optimal and robust.

| weight for $\mathcal{L}_{\text{forecast}}$ | 0.2 | 0.5 | 1.0 | 1.2 | 1.5 | 1.8 |
|---|---|---|---|---|---|---|
| Accuracy (%) | 63.5 | 64.3 | **64.6** | **64.6** | **64.6** | 63.1 |

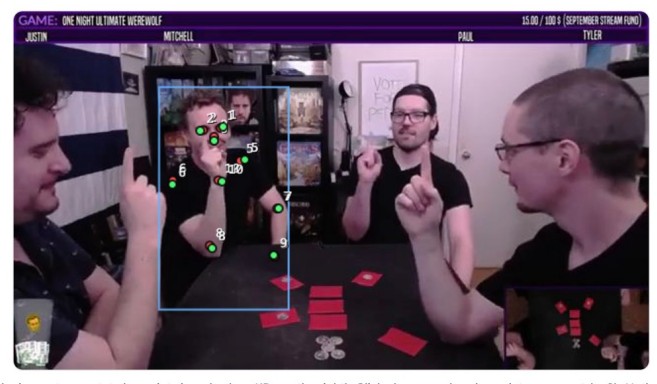 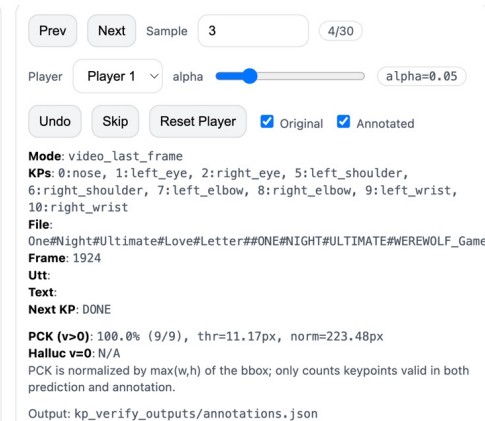

Figure 12: Illustration of web-based keypoint verification tool for sampled video frames.

## A.7 Societal Impacts and Concerns

Although Online-MMSI has the potential to support assistive and collaborative applications, it also introduces societal risks. In particular, fine-grained recognition of gaze, gesture, and conversational dynamics could be repurposed for surveillance of social interactions, workplace monitoring, or targeted manipulation. Such uses raise privacy and ethical concerns, especially if deployed without consent or oversight. Moreover, errors in speech recognition, tracking, or gaze estimation may disproportionately affect certain groups, amplifying bias and inequity. These considerations highlight the importance of responsible deployment, including transparency, privacy-preserving design, and appropriate governance mechanisms.

## A.8 Human Validation on Annotations

To facilitate the socially-aware visual prompting, we reuse the AlphaPose-based annotations released by Lee et al. (2024a), including identity-tracked bounding boxes and upper-body keypoints, rather than generating new annotations in this work. These annotations are produced by AlphaPose and further curated using human-verified reference frames to align verbal and non-verbal cues. Specifically, AlphaPose outputs may occasionally suffer from missing detections, for example when a participant is temporarily out of the camera view. In such cases, we leverage human-verified reference player positions provided by Lee et al. (2024a) to correct missing player locations and maintain consistent identity tracking.

To further assess annotation quality, we conducted an additional lightweight human validation on per-joint pose accuracy. Using a web-based relabeling tool as shown in Fig. 12, we randomly sampled 30 video clips and manually reviewed all upper-body keypoints (eyes, nose, shoulders, elbows, and wrists) for every detected player in the last frame. This resulted in 129 player entries and a total of 1,161 keypoints. For visible keypoints, we obtain a bounding-box-normalized Percentage of Correct Keypoints (PCK) at $\alpha = 0.05$ of 0.912. Keypoints marked as not visible are excluded from the PCK computation, as they lack uniquely verifiable ground-truth locations. Overall, this human validation indicates that the automated annotations are generally accurate for observable joints and suitable for socially-aware visual prompting.

Table 11: Ablation study on different pose estimation models evaluated on the YouTube subset. Results are reported as accuracy (%). Our method is robust to the specific pose estimation backbone.

| Pose Model | STI | PCR | MPP | Avg. Accuracy |
|---|---|---|---|---|
| AlphaPose | 64.58 | 68.83 | 47.20 | 60.20 |
| YOLO-Pose | 64.13 | 68.54 | 47.61 | 60.09 |

### A.9 Details on Instruction-Tuning Data Construction

We construct instruction-tuning data specifically for the Online-MMSI setting, where models are required to perform multimodal social reasoning using only historical information. Online-MMSI emphasizes short-horizon social dynamics, in which a speaker's referent is typically resolved through recent conversational turns. Each instruction-tuning instance is constructed using a fixed-length sliding window over the most recent $d = 10$ dialogue turns. This window corresponds to an average temporal span of approximately 28 seconds and is sufficient to capture the local conversational and social cues required for referent resolution in online settings. Each training instance is formatted as a standard user-assistant pair. As shown in Fig. 13, the user input concatenates the historical multimodal context, including annotated video frames and dialogue history, with the task-specific instruction. The assistant output, as shown in Fig. 14, contains both (i) the task answer (i.e., the resolved referent) and (ii) the multi-party conversation forecasting targets, which include upcoming speaker turns and their corresponding utterances.

> **User**
>
> "In the video, the red, blue, green, yellow, purple, and orange colors correspond to Player0, Player1, Player2, Player3, Player4, and Player5, respectively. The transcripts are: [Player2]: All right. What was your point, if she was a Werewolf? [Player0]: Well, she would want everybody to kill her if she was a Tanner, but she doesn't want anybody kill her. So, I think she's a werewolf. [Player2]: Okay. [Player0]: Because she's trying to say that you guys are the. [Player2]: Okay. [Player0]: And she put the Tanner out there. [Player2]: Okay. [Player0]: So, I think she's trying to deflect and calls everybody to vote for her. [Player1]: Because, [MASK]'s saying he was a Robber. Determine which player a pronoun refers to in the position of [MASK]. Predict the upcoming speakers' turns. Predict the upcoming conversations."

Figure 13: An example of user input prompt for Online-MMSI instruction.

> **Assistant**
>
> "Player3. Player0, Player1, Player3, Player1. [Player0]: I mean, calls everybody to vote for you guys. [Player1]: Let's say he robbed you and the Insomniac. You're not disagreeing with him, but you would've looked your card after he robbed you. So the question is, were you a Robber when... [Player3]: The question is, was I lying or were you lying? [Player1]: Were you lying? One of you is a liar, or you're both Werewolves."

Figure 14: An example of assistant output for Online-MMSI instruction.

### A.10 Ablation Studies on Pose Estimation Models

To evaluate the influence of different pose estimation models, we replace AlphaPose with YOLO-Pose (Maji et al., 2022) and conduct an ablation study. Specifically, we reuse the tracking results and bounding boxes released by Lee et al. (2024a), and estimate upper-body keypoints for each participant using YOLO-Pose. As shown in Table 11, the choice of pose estimator has minimal impact on performance. The average accuracy differs by only 0.11%, indicating that our method is robust to the specific pose estimation backbone.

### A.11 Ablation Studies on Larger Backbone Models

We further evaluate the impact of scaling backbone model size by conducting experiments with larger vision–language models, Qwen2.5-VL-32B and Qwen3-VL-32B, on the YouTube subset across three social

Table 12: Ablation study on larger backbone models evaluated on the YouTube subset. Results are reported as accuracy (%). Larger backbone models consistently outperform their smaller counterparts, indicating that increased model capacity benefits online multi-modal social reasoning.

| Model | STI | PCR | MPP | Avg. Accuracy |
|---|---|---|---|---|
| Qwen2.5-VL-7B | 64.58 | 68.83 | 47.20 | 60.20 |
| Qwen2.5-VL-32B | 66.26 | **69.41** | **49.93** | **61.87** |
| Qwen3-VL-32B | **67.63** | 67.68 | 49.65 | 61.65 |

interaction tasks. As shown in Table 12, larger backbone models consistently outperform their smaller counterparts, indicating that increased model capacity benefits Online-MMSI. Specifically, Qwen2.5-VL-32B improves the average accuracy by 1.67% over Qwen2.5-VL-7B, while Qwen3-VL-32B achieves a 1.45% gain.

### A.12 Additional Qualitative Comparison

Fig. 15 presents additional qualitative comparisons between our method and representative baselines (i.e., Lee et al. (2024a) and Vanilla Qwen) across the three social interaction tasks. Overall, our approach more reliably infers referents from ongoing conversation turns, which we attribute to the effects of multi-party conversation forecasting and socially-aware visual prompting in capturing complex social dynamics. In the first STI example, when Player4 turns his head to ask Player3, "Are you a werewolf?", our method successfully captures the head motion and correctly identifies the speaking target. In the second PCR example, after Player2 states "I was a seer. Oh yeah." and the discussion continues for several subsequent turns, our model correctly infers that Player1 is referring to Player2, whereas the baseline methods fail to maintain this referential consistency. In the third MPP example, Player1 asks multiple participants the same question, "Who are you voting for?"; our approach correctly predicts that the mentioned player is Player4 by jointly leveraging gaze direction and conversational context. We further observe that Lee et al. (2024a) incorrectly predicts the speaker as the referent in both the second and third examples. This behavior is likely due to limitations of its language backbone (BERT (Devlin et al., 2019)), which struggles to disentangle speaker identity from referential targets when referent identification becomes challenging.

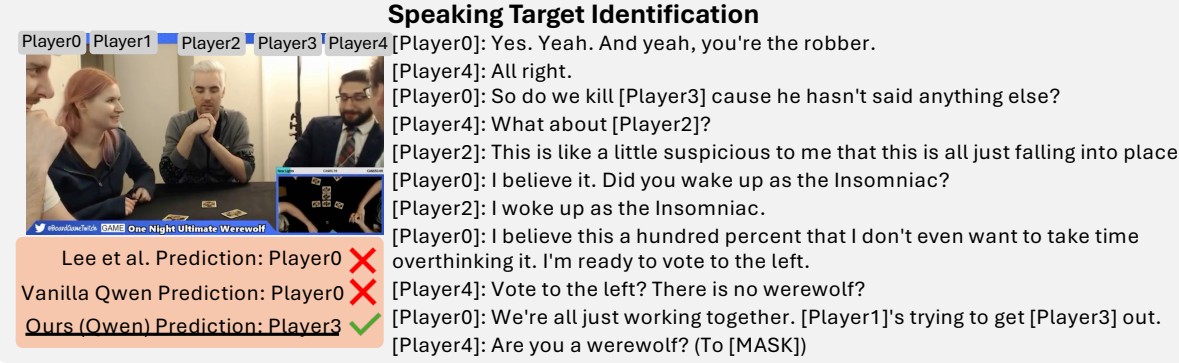

**Speaking Target Identification**

[Player0]: Yes. Yeah. And yeah, you're the robber.
[Player4]: All right.
[Player0]: So do we kill [Player3] cause he hasn't said anything else?
[Player4]: What about [Player2]?
[Player2]: This is like a little suspicious to me that this is all just falling into place.
[Player0]: I believe it. Did you wake up as the Insomniac?
[Player2]: I woke up as the Insomniac.
[Player0]: I believe this a hundred percent that I don't even want to take time overthinking it. I'm ready to vote to the left.
[Player4]: Vote to the left? There is no werewolf?
[Player0]: We're all just working together. [Player1]'s trying to get [Player3] out.
[Player4]: Are you a werewolf? (To [MASK])

Lee et al. Prediction: Player0 ✗
Vanilla Qwen Prediction: Player0 ✗
Ours (Qwen) Prediction: Player3 ✓

**Pronoun Coreference Resolution**

[Player2]: So, here's my thing is that hers a wild card.
[Player0]: Okay.
[Player3]: I agree with this, but you were-
[Player2]: I was a seer. Oh yeah.
[Player3]: A hundred percent seer? I was-
[Player2]: Did I forget to look? Yes.
[Player1]: I honestly think we're good this game.
[Player0]: I think so.
[Player1]: What makes you trust her?
[Player0]: That looks sketchy.
[Player1]: What makes me trust her is that she came out and was very confident that [MASK] (originally she) was herself even after I said there's a troublemaker.

Lee et al. Prediction: Player1 ✗
Vanilla Qwen Prediction: Player0 ✗
Ours (Qwen) Prediction: Player2 ✓

**Mentioned Player Prediction**

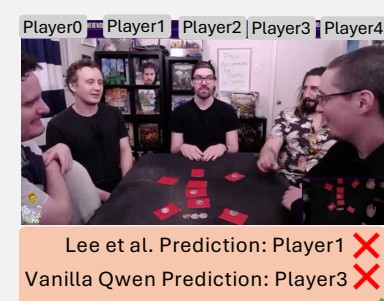

[Player3]: Okay, so...
[Player2]: Who are you going to vote for?
[Player3]: Hold on. He saw a werewolf ...
[Player2]: I'll tell [Player3] at the last second.
[Player3]: ... in the center.
[Player2]: That's really like rude.
[Player3]: No, no, no.
[Player1]: Who are you voting for?
[Player2]: I don't know. I need some help.
[Player3]: Yeah, yeah, yeah. We're in trouble.
[Player1]: [MASK] (originally [Player4]), who are you voting for?

Lee et al. Prediction: Player1 ✗
Vanilla Qwen Prediction: Player3 ✗
Ours (Qwen) Prediction: Player4 ✓

Figure 15: Qualitative comparison illustrating the advantages of multi-party conversation forecasting and socially-aware visual prompting for online social interaction understanding tasks.

