# OpenReview forum: "Towards Online Multimodal Social Interaction Understanding"
_TMLR — Accepted by TMLR_

### Review · Reviewer_ERFD · 2025-09-01

**Summary Of Contributions:**

This paper introduces a new problem **Online Multimodal Social Interaction Understanding (Online-MMSI)**, which requires AI models to interpret social interactions using only historical video and dialogue context, without access to future information.

The authors propose **Online-MMSI-VLM,** which using Conversation forecasting and Visual Prompting techniques to enrich contextual understanding and focus on relevant visual context. Results show that Online-MMSI-VLM outperforms baselines and even humans in online settings, demonstrating the effectiveness of the proposed components.

**Audience:**

Yes

**Audience Explanation:**

This work propose a new problem which is valuable for MMSI and real-time AI applications.

The MMSI community will be interested in this paper. And the proposed question is valuable in practice.

**Claims And Evidence:**

Yes

**Claims Explanation:**

- The experimental design is extensive, with multiple tasks, two datasets, and extensive ablation studies.
- The latency of model is reported, strengthening the paper's claims about its suitability for real-time deployment

**Requested Changes:**

- Generalization. The VLM is trained and tested on same datset with the same domain (social deduction games). It would be valid to show the generationzation between datasets and between domain (like online meetings) .
- Limited Baselines. Only one main baseline is considered in this work.  it would be beneficial to compare against other capable MLLMs or streaming video understanding models.
- End-to-end latency The authors only report the latency of generation. What is the whol latency including processing speech, do forecating and generating answer for a 11B model? This would offer a more complete picture of the system's feasibility in a true online scenario.

---

### Review · Reviewer_QsZG · 2025-09-08

**Summary Of Contributions:**

The paper proposes the Online-MMSI for multimodal social interaction understanding based on predictive modeling characterizing speaker changes (speaker turns) and utterances (content) as well as modeling accounting for visual cues in a time-causal (online) setting (i.e., only past and current data used to predict future social interaction). The approach uses Llama and Qwen with instruction tuning based on LoRa whereas AlphaPose is used to extract visual annotations. Data and evaluation design is taken from Lee et al 2024a and the contribution of the paper is thus to extend this framework to the online setting. The results show that the proposed approach is superior to the prior work of Lee et al. 2024 in the online setting (which lee et al. 2024a is not trained to handle) and model ablations confirm the benefits of the various introduced design choices including both MMSI and forecasting in training, conversation forecasting length and visual prompting employed.

Strenghts:
* The paper advances multimodal social interaction modeling to the online setting whereas past work (Lee et al 2024a) were designed for the offline setting.
* Results appear to improve upon alternative existing modeling strategies.

Weaknesses:
* It appears the comparison to Lee et al 2024a is for models trained in the off-line setting and not online setting.
* The methodological innovations are limited mainly based on combining existing tools.

**Audience:**

Yes

**Audience Explanation:**

Whereas prior work has considered multimodal social interaction modeling from an off-line setting (i.e., Lee et al 2024a) the current paper advances such modeling to the online setting which is a valid and relevant contribution. The methods contribution is somewhat incremental utilizing and combining existing tools such as Llama and Qwen instruction tuned by LoRa and with AlphaPose for visual annotations extraction. Data and evaluation design are largely also taken from Lee et al 2024a but adopted to the online setting. Overall this makes the nature of the paper somewhat incremental.

**Broader Impact Concerns:**

While the paper describes positive aspects of social aware AI assistants for social support and collaborative tasks, the paper should also discuss negative societal impacts and concerns such as algorithms for surveillance purposes of social interactions etc.

**Claims And Evidence:**

Yes

**Claims Explanation:**

The approach is contrasted Lee et al 2024a, however, it appears this is designed to only be trained in a non-causal setting (not online setting) and a fair comparison would require Lee et al. 2024a be adapted to an online setting (which is described in the introduction as feasible).  It is very unclear if this is the case but if trained in the traditional non-online manner and compared in the online setting this would not be a very fair comparison. It would then be relevant to see how well the proposed method compare in online performance to Lee et al. 2024a also trained in an online setting (as discussed at the end of section 2.1). It is also unclear why the results in the paper are not compared to GPT-4o and Gemini2.5Pro in Table 1 as provided results for in Figure 1. It would be nice to include such comparisons in the table although these methods provide subpar performance.

That being said the experimental design is sound and valid and the approach provides good results with reasonable systematic ablations evidencing the benefit of various design choices.

**Requested Changes:**

Compare to Gemini2.5Pro and GPT-4o for the results in Table 1 and include Lee et al 2024a both trained in an off-line and on-line setting for comparison as discussed possible end of section 2.1.

In Figure 1 why is Player2 denoted Player2Player3 in the top left figure?
-	The description of what  “Prior model” in Figure 1 (2) refers to should also be clarified, i.e. Lee et al. 2024a?

How are the two objectives L_MMSI and L_forecast balanced and should there here be weights tuned to optimally combine the two terms?

Please clarify why the different step sizes respectively for speaker identification and mentioned player as opposed to pronoun coreference tasks are used and not kept the same step-size for all three learning tasks.

The paper would benefit from a bit of careful proof-reading:
The VLM abbreviation is never introduced (Visual Language Model) – please explain this abbreviation at first occurrence.
which benefit addressing dynamic multimodal > which makes them useful for dynamic multimodal
Furthure analysis -> Further analysis

---

### Review · Reviewer_uLVX · 2025-12-24

**Summary Of Contributions:**

**Summary**

Based on Multimodal Social Interaction Understanding (MMSI), this paper introduces Online-MMS, which aims to identify social cues (speaking target, coreference, and mentioned player) using only historical information, thereby prohibiting access to future context, defined as an “online” setting. The authors trimmed future frames and transcripts from the existing Werewolf Among Us dataset to build this task. The authors also propose Online-MMSI-VLM, which leverages AlphaPose-generated bounding boxes and keypoints to construct instructions to tune MLLMs. The proposed method is compared against baselines like InternVL, Qwen, and LLaMA, achieving SoTA performance in this specific online setting.

**Strengths**

1.	The idea of this paper is intuitive and easy to understand.
2.	Developing a real-time AI assistant for MMSI or Egocentric (i.e., AR Glasses) is a challenge and critical research direction.
3.	It is interesting to evaluate models in a streaming-like setup, which is more aligned with real-world deployment than static benchmarks.


**Weaknesses**

1.	Overlap with existing egocentric tasks. The core contribution of leveraging historical context for online understanding overlaps significantly with existing research in Egocentric AI Assistants [1, 2, 3]. Tasks such as EgoLife’s EventRecall [1] and Ego4D’s NLQ [2] already utilize historical video context to answer user questions or provide assistance in real-time. Specifically, works like EgoLife [1] have already explored using historical information for instruction-tuning GPT models for long-context video QA, which the authors overlook. This lack of comparison makes me concern the novelty and necessity of defining "Online-MMSI" as a distinct new task.

2.	Unicentric task definitions. The definition of the subtasks in Section 3 is overly simplistic and lacks self-containment. The authors merely list the tasks (i.e., "speaking target identification," "pronoun coreference resolution," and "mentioned player prediction") without defining the specific inputs, outputs, or success criteria, effectively forcing the reader to consult external sources [4] to understand the basic premise. As a submission to a journal, the paper should be self-contained and clearly define the problem space within the text.

3.	Limited dataset contribution. The proposed dataset appears to be merely a subset of an existing dataset with truncated timestamps, which limits its novelty. Furthermore, the new annotations rely on AlphaPose to generate bounding boxes and keypoints automatically. The paper lacks a detailed description of quality control, human verification of these automated annotations, or metrics assessing their accuracy in this specific domain. This raises me concern the data quality and its ultimate value to the community.

4.	Lack of technology contribution. The primary technical contribution is the application of SFT with LoRA on an existing MLLM. As this is a standard setting in current research, I am concerned that the technical contribution may not be sufficient for the TMLR community without further methodological innovation.

5.	Missing efficiency verification for the “Online” Claims. The paper frames the work as an "Online" task aiming for "immediate feedback," implying a requirement for real-time application. However, the authors overlook a critical efficiency analysis. Efficiency is a major challenge for online tasks, yet there is no comparison of latency or memory consumption against API-based MLLMs or other open-source baselines. I recommend including a table comparing the inference speed and resource usage of the proposed method against baselines to verify that "immediate feedback" is achievable.

6.	Lack of details on instruction tuning. The paper lacks sufficient detail regarding the construction of the instruction tuning data. It is unclear how the historical context was sampled or formatted for the SFT process. It is a recommendation to proposed sample data in your supplementary.

7.	Insufficient ablation studies. For example, What is the necessity and impact of the specific modules in Socially-aware Visual Prompting (e.g., performance without bounding boxes vs. without upper body keypoints)? What is the influence of using different pose estimation models other than AlphaPose? What is the performance of larger backbone models (e.g., Qwen3-VL-32B)?

References

[1] Grauman K, Westbury A, Byrne E, et al. Ego4d: Around the world in 3,000 hours of egocentric video[C]//Proceedings of the IEEE/CVF conference on computer vision and pattern recognition. 2022: 18995-19012.

[1] Yang J, Liu S, Guo H, et al. Egolife: Towards egocentric life assistant[C]//Proceedings of the Computer Vision and Pattern Recognition Conference. 2025: 28885-28900.

[2] Damen D, Doughty H, Farinella G M, et al. The epic-kitchens dataset: Collection, challenges and baselines[J]. IEEE Transactions on Pattern Analysis and Machine Intelligence, 2020, 43(11): 4125-4141.

[4]  Lee S, Lai B, Ryan F, et al. Modeling multimodal social interactions: new challenges and baselines with densely aligned representations[C]//Proceedings of the IEEE/CVF Conference on Computer Vision and Pattern Recognition. 2024: 14585-14595.

**Additional Comments:**

Please refer to the Weakness section.

**Audience:**

Yes

**Audience Explanation:**

The general direction of the paper is relevant to the community. Specifically,

1. Relevance. Developing real-time/streaming AI assistants and evaluating them in a dynamic setup (as opposed to static benchmarks) is a challenging and practical problem that aligns with real-world deployment needs (see Strengths 2 & 3).

2. Caveat. While the topic is interesting, the audience's interest might be dampened by the significant overlap with existing egocentric tasks like EgoLife and Ego4D. The authors need to better clarify the unique value of "Online-MMSI" to distinguish it from these well-known retrospective tasks (see W.1). However, the streaming evaluation perspective itself holds value.

**Claims And Evidence:**

Yes

**Claims Explanation:**

The submission required more evidence to support it claims, particularly regarding the "Online/Real-time" nature of the task and the quality of the dataset. Specifically,

1. Missing evidence for "Online" capabilities. The paper frames the method as an "Online" assistant for immediate feedback, but fails to provide any efficiency analysis (e.g., latency, memory consumption) to prove it can operate in a real-time streaming setup (see Weakness 5). Without this, the claim of "online" applicability is unsupported.

2. Dataset reliability. The proposed dataset relies heavily on automated annotations (AlphaPose) without any reported quality control or human verification. There is no metric provided to validate the accuracy of these ground truths, which weakens the empirical results (see W.3).

3. Ambiguous Definitions. The subtasks are listed without formal inputs/outputs or success criteria, making the evaluation protocol unclear and difficult to reproduce (see W.2).

4. Insufficient Ablation. Critical design choices (e.g., specific modules in visual prompting) lack ablation studies to justify their necessity (see W.7).

**Requested Changes:**

Major:

1.	Clarify the novelty and contribution relative to overlapping history-based Egocentric AI tasks (i.e., Ego4D’s NLQ and Egolife’s EventRecall).

2.	Provide self-contained definitions for the three subtasks in Section 3.

3.	Add quantitative efficiency experiments (latency, memory) to verify the "online" and "immediate feedback" claims against baselines.
4.	Provide validation or human evaluation for the automated annotations to ensure dataset quality.

5.	Provide more details on the instruction tuning data construction, specifically the sampling and formatting of historical context.

6.	Expand ablation studies. Please refer to W.7.

Minor:

1.	It is recommended to provide more qualitative experiments to compare the baseline models.

2.	It is recommended to provide common failure cases

---

> ### Author Response · Authors · 2026-01-07
> **Response to Reviewer uLVX**
>
> We sincerely thank the Reviewer uLVX for the time and effort in reviewing the paper and providing valuable comments! As per the reviewer's suggestions, we have revised our manuscript, and the updated parts are highlighted using blue color. We are happy to make any further changes as requested.
>
> 1. Clarify the novelty and contribution relative to overlapping history-based Egocentric AI tasks (i.e., Ego4D’s NLQ [1] and Egolife’s EventRecall [2]).
>
>    - (Response): Thanks for mentioning these papers. We have added an explicit discussion in Section 3 of the main paper to clarify the relationship and distinctions between Online-MMSI and history-based egocentric AI tasks. While both lines of work leverage historical video context, Online-MMSI differs from egocentric tasks along the following three dimensions:
>
>    - Social interaction understanding vs. retrospective event retrieval and grounding. Online-MMSI targets high-level social interaction understanding: given a multi-party conversation and video stream, the model must infer who is being referred to in the current utterance by jointly leveraging verbal cues (speaker turns, conversational dynamics) and non-verbal cues (gaze, posture, gestures). In contrast, retrospective egocentric tasks such as Ego4D NLQ and EgoLife EventRecall are formulated as retrospective event retrieval and grounding problems over long egocentric video, answering queries like “when/where did X occur?” or “what happened?”.
>
>    - Short-horizon online situation modeling vs. long-horizon online memory retrieval. Online-MMSI inherently focuses on short-horizon situation modeling, as a speaker’s referent is typically determined by local social dynamics while information from long-term history quickly becomes irrelevant for the current utterance. In contrast, history-based egocentric tasks such as Ego4D’s NLQ and EgoLife’s EventRecall emphasize long-horizon online memory retrieval, where the system accesses, compresses, and queries extended video histories to retrieve information that may have occurred far in the past (e.g., tens of minutes or hours earlier).
>
>    - Identity consistency and tracking requirements. Online-MMSI requires consistent identity tracking of multiple participants (i.e., maintaining “who is who” over time) to correctly resolve referential targets. Online-MMSI videos typically assumes a relatively stable third-person or fixed-view setting, where participants remain visible across consecutive frames. In contrast, egocentric videos are captured from a first-person, moving camera, where rapid viewpoint changes, head motion, and frequent scene transitions make persistent multi-person identity tracking unreliable or ill-defined. As a result, Ego4D’s NLQ and EgoLife’s EventRecall generally do not require maintaining identity consistency across multiple people as a primary output variable.
>
>
> 2. Provide self-contained definitions for the three subtasks in Section 3.
>
>    - (Response): Thanks for the suggestion! We have revised Section 3 in the main paper to provide self-contained definitions for the three subtasks.
>
>    - Speaking Target Identification (STI). This task aims to identify who the current speaker is talking to, given historical context where the current utterance contains a second-person reference (e.g., "you", "your").
>
>    - Pronoun Coreference Resolution (PCR). This task aims to determine who the current pronoun refers to, given historical context where the current utterance contains a third-person pronoun (e.g., "he", "she", "him", "her", "his").
>
>    - Mentioned Player Prediction (MPP). This task aims to predict who is referred to by their name, given historical context where the participant name (e.g., "Mike") in current utterance is masked.
>
>    - The task response should be the identity of one participant, i.e. "Player0", "Player1", ..., or "PlayerN". A prediction is considered correct if the output identity matches the ground truth referent.

---

> ### Author Response · Authors · 2026-01-07
> **Response to Reviewer uLVX (continued)**
>
> 3. Add quantitative efficiency experiments (latency, memory) to verify the "online" and "immediate feedback" claims against baselines.
>
>    - (Response): Thanks! Appendix A.6 previously reported the end-to-end latency of Online-MMSI-VLM (Qwen) (≈343 ms), supporting the immediate-feedback claim. For completeness, we additionally include efficiency evaluations (i.e. latency and memory) on representative baselines, and relocate the revised analysis from Appendix A.6 to Section 5.8 of the main paper.
>
>    - We compare the inference latency of different baselines. Online-MMSI-VLM maintains sub-second latency, while API-based methods and Lee et al. (2024a) exhibit higher response times. The increased latency of Lee et al. (2024a) is largely attributable to its offline processing pipeline, which leverages future context and therefore introduces inherent delays.
>
>     | Model                     | Latency (s) |
>     |---------------------------|-------------|
>     | Lee et al. (2024a)        | 14.010      |
>     | GPT-4o                    | 3.846       |
>     | Gemini 2.5 Pro            | 2.207       |
>     | Vanilla Intern VL         | 2.953       |
>     | Vanilla Qwen              | 0.268       |
>     | Vanilla Llama             | 0.533       |
>     | Online-MMSI-VLM (Qwen)    | 0.302       |
>     | Online-MMSI-VLM (Llama)   | 0.554       |
>     Table: Latency Evaluation
>
>    - We additionally evaluate the GPU memory occupation during inference for open-source large language models. Online-MMSI-VLM requires approximately 20 GB of GPU memory at inference time. This memory footprint is well within the capacity of widely adopted embedded computing platforms such as Jetson AGX Orin, which provides 32~GB unified memory, thereby enabling stable real-time online inference in practical deployment settings.
>
>     | Model                  | GPU Memory (GB) |
>     |------------------------|-----------------|
>     | Vanilla Intern VL      | 21.31           |
>     | Vanilla Qwen           | 17.26           |
>     | Vanilla Llama          | 21.49           |
>     | Online-MMSI-VLM (Qwen) | 17.39           |
>     | Online-MMSI-VLM (Llama)| 21.52           |
>     Table: Memory Consumption Evaluation
>
> 4. Provide validation or human evaluation for the automated annotations to ensure dataset quality.
>
>     - (Response): Thanks! We respectfully clarify that we do not claim the dataset itself as a primary contribution of this work. As stated in the introduction of the main paper, our contributions are threefold:
>     - We propose a new task, Online-MMSI, where the AI assistant must interpret multimodal social interactions and provide immediate feedback using only historical dialogues and videos. 2) We propose Online-MMSI-VLM, a multimodal large language model-based framework that integrates multi-party conversation forecasting and socially-aware visual prompting. To the best of our knowledge, it is the first work to use multimodal large language models for advancing MMSI. 3) Extensive experiments, for three tasks on two social datasets, demonstrate the effectiveness of our approach and show Online-MMSI-VLM establishes a strong benchmark for Online-MMSI.
>     - The dataset is originally from Lee et al. (2024a), but we propose a novel online setting and restructure the data for this online problem. Following their practice, we reuse the AlphaPose-based annotations, including identity-tracked bounding box and keypoints, released by Lee et al. (2024a), rather than generating new annotations in this work. Indeed, AlphaPose outputs might have issues of player missing detection, e.g., a player is temporarily undetected due to out of screen. In this situation, human verified reference player position is leveraged to correct the missing player position.
>     - To better address the reviewer’s concern, we conducted an additional lightweight quality check on per-joint pose accuracy. Using a web-based relabeling tool, we randomly sampled 30 clips and manually reviewed all upper-body keypoints (eyes, nose, shoulders, elbows, and wrists) for every detected player in the last frame (including 129 player entries with 1,161 keypoints). On visible keypoints, we obtain a bbox-normalized Percentage of Correct Keypoints (PCK) at α = 0.05 of 0.912. Note that keypoints marked as not visible are excluded from PCK computation, as they lack uniquely verifiable ground-truth locations and would render the metric ill-posed. Overall, this human validation demonstrates that the automated annotations are generally accurate for observable joints. We have included the keypoint verification in Appendix A.8.

---

> ### Author Response · Authors · 2026-01-07
> **Response to Reviewer uLVX (continued)**
>
> 5. Provide more details on the instruction tuning data construction, specifically the sampling and formatting of historical context.
>
>     - (Response): Nice suggestion! We have added detailed descriptions of the instruction-tuning data in Appendix A.9 and included representative instruction-tuning examples in the supplementary material.
>     - Historical context sampling. Unlike prior history-based tasks that rely on long-range context (e.g., event recall or temporal localization), Online-MMSI focuses on short-horizon social dynamics, where a speaker’s referent is typically resolved through recent conversational turns. Accordingly, we construct each instruction-tuning instance using a fixed-length sliding window of the most recent d=10 dialogue turns, which corresponds to an average temporal span of approximately 28 seconds.
>     - Instruction formatting. Each training instance is formatted as a standard user–assistant pair. The user input concatenates the historical multimodal context with the task instruction, while the assistant output includes both the task answer and the conversation forecasting targets. An example is shown below:
>     - "user": "In the video, the red, blue, green, yellow, purple, and orange colors correspond to Player0, Player1, Player2, Player3, Player4, and Player5, respectively. The transcripts are: [Player2]: All right. What was your point about, if she was a Werewolf? [Player0]: Well, she would want everybody to kill her if she was a Tanner, but she doesn't want anybody kill her. So, I think she's a werewolf. [Player2]: Okay. [Player0]: Because she's trying to say that you guys are the... [Player2]: Okay. [Player0]: And she put the Tanner out there. [Player2]: Okay. [Player0]: So, I think she's trying to deflect and calls everybody to vote for her. [Player1]: Because, [MASK]'s saying he was a Robber. Determine which player a pronoun refers to in the position of [MASK]? Predict the upcomping speakers' turns. Predict the upcoming conversations."
>     - "assistant": "Player3. Player0, Player1, Player3, Player1. [Player0]: I mean, calls everybody to vote for you guys. [Player1]: Let's say he robbed you and the Insomniac. You're not disagreeing with him, but you would've looked your card after he robbed you. So the question is, were you a Robber when... [Player3]: The question is, was I lying or were you lying? [Player1]: Were you lying? One of you is a liar so or you're both Werewolves."
>
> 6. What is the necessity and impact of the specific modules in socially-aware visual prompting (e.g., performance without bounding boxes vs. without upper body keypoints)?
>
>     -  (Response): Great suggestion! We have revised Table 5 and Section 5.5 to more clearly illustrate the impact and necessity of each visual prompting module.
>
>     - As shown in the table, incorporating bounding boxes (BB) and keypoints (KP) improves average accuracy over raw RGB frames by 1.1% and 0.72% respectively, and further combining bounding boxes and keypoints yields the best results. It indicates that bounding boxes and keypoints contribute complementary social cues at different levels of granularity. Bounding boxes primarily highlight coarse-level information, such as participants’ spatial location, body orientation, and relative positioning. In contrast, upper-body keypoints emphasize finer-grained signals, including gaze direction and hand gestures.
>
>    | Visual Input        | Speaking Target Identification | Pronoun Coreference Resolution | Mentioned Player Prediction | Avg. Accuracy |
>    |--------------------|-------------------------------|-------------------------------|-----------------------------|---------------|
>    | RGB                | 63.96                         | 65.20                         | 46.52                       | 58.56         |
>    | RGB + BB           | 64.13                         | 68.13                         | 46.71                       | 59.66         |
>    | RGB + KP           | 63.96                         | 66.79                         | 47.10                       | 59.28         |
>    | RGB + BB + KP      | **64.58**                         | **68.83**                         | **47.20**                       | **60.20**         |

---

> ### Author Response · Authors · 2026-01-07
> **Response to Reviewer uLVX (continued)**
>
> 7. What is the influence of using different pose estimation models other than AlphaPose?
>    - (Response): Thanks. We have included the ablation in Appendix A.10. To evaluate the influence of different pose estimation models, we replace AlphaPose with YOLO-Pose [3] and conduct an ablation study. Specifically, we reuse the tracking results and bounding boxes released by Lee et al. (2024a), and estimate upper-body keypoints for each participant using YOLO-Pose. As shown in the table, the choice of pose estimator has minimal impact on performance. The average accuracy differs by only 0.11%, indicating that our method is robust to the specific pose estimation backbone.
>
>     | Pose Estimator | Speaking Target Identification | Pronoun Coreference Resolution | Mentioned Player Prediction | Avg. Accuracy |
>     |----------------|-------------------------------|-------------------------------|-----------------------------|---------------|
>     | AlphaPose      | **64.58**                         | **68.83**                         | 47.20                       | **60.20**         |
>     | YOLO-Pose      | 64.13                         | 68.54                         | **47.61**                       | 60.09         |
>
> 8. What is the performance of larger backbone models (e.g., Qwen3-VL-32B)?
>
>     - (Response): Thanks. We have included the ablation in Appendix A.11. We further evaluate the impact of scaling backbone model size by conducting experiments with larger vision–language models, Qwen2.5-VL-32B and Qwen3-VL-32B, on the YouTube subset across three social interaction tasks. As shown in Table, larger backbone models consistently outperform their smaller counterparts, indicating that increased model capacity benefits Online-MMSI. Specifically, Qwen2.5-VL-32B improves the average accuracy by 1.67% over Qwen2.5-VL-7B, while Qwen3-VL-32B achieves a 1.45% gain.
>
>     | Model            | Speaking Target Identification | Pronoun Coreference Resolution | Mentioned Player Prediction | Avg. Accuracy |
>     |------------------|-------------------------------|-------------------------------|-----------------------------|---------------|
>     | Qwen2.5-VL-7B    | 64.58                         | 68.83                         | 47.20                       | 60.20         |
>     | Qwen2.5-VL-32B   | 66.26                         | **69.41**                         | **49.93**                       | **61.87**         |
>     | Qwen3-VL-32B     | **67.63**                         | 67.68                         | 49.65                       | 61.65         |
>
>
> 9. Provide more qualitative experiments to compare the baseline models.
>    - (Response): Thanks. We have added more qualitative comparison with baselines in Appendix A.12. Overall, our approach more reliably infers referents from ongoing conversation turns, which we attribute to the effects of multi-party conversation forecasting and socially-aware visual prompting in capturing complex social dynamics.
>
> 10. Provide common failure cases.
>     - (Response): Nice suggestion. We have added common failure cases in Section 5.9. We observe that the model is likely to fail when social dynamics overlap or when social interaction cues are weak or ambiguous. E.g., two participants (Player2 and Player3) simultaneously address different interlocutors, creating ambiguity in addressee resolution.
>
> 11. Technology contribution.
>     - (Response): Thanks for raising this concerns. We agree that our implementation adopts standard supervised instruction tuning with LoRA. However, we respectfully argue that this is not the core technical contribution of our work. Instead, We propose a novel task, Online-MMSI. It aims to address "Who is being referred to in current utterance?", which requires modeling current social status by leveraging history information and compensating for missing future context.
>     - To address Online-MMSI, we made technical contribution in reasoning structure: (i) multi-party conversation forecasting as an auxiliary reasoning objective to compensate for missing future context, and (ii) socially-aware visual prompting to explicitly highlight visual social cues (e.g., posture, gaze, gesture) that become critical in online setting. The experiments show that these components provide consistent and complementary gains compared to standard SFT pipeline.
>
> Thank you again for your valuable feedback. We welcome any further questions or suggestions you may have.
>
> [1] Grauman, Kristen, et al. "Ego4d: Around the world in 3,000 hours of egocentric video." CVPR. 2022.
>
> [2] Yang, Jingkang, et al. "Egolife: Towards egocentric life assistant." CVPR. 2025.
>
> [3] Maji, Debapriya, et al. "Yolo-pose: Enhancing yolo for multi person pose estimation using object keypoint similarity loss." CVPR. 2022.

---

> > ### Comment · Action_Editor_yAh4 · 2026-02-11
> >
> > Dear reviewer uLVX,
> >
> > Did the authors' response address your concern? Can you post your final recommendation? Thanks!

---

### Decision · Action_Editor_yAh4 · 2026-02-25

**Recommendation:** Accept as is

**Audience:**

Yes

**Audience Explanation:**

The reviewers unanimously agree that the TMLR audience would benefit from the introduction of the Online Multimodal Social Interaction (Online-MMSI) problem. This framework is considered practically relevant and well-motivated for researchers focused on real-time AI interpretation of social dynamics and historical context. While the reviewers are not very enthusiastic about the technical novelty, they like the successful integration of conversation forecasting and visual prompting in a social deduction setting. The work contributes a relevant, high-performing approach to the growing field of multimodal social understanding.

**Claims And Evidence:**

Yes

**Claims Explanation:**

After the rebuttal, all three reviewers unanimously agree that the claims are well-supported following the authors' revisions and the addition of comprehensive ablation studies. Specifically:
1. Human verification of annotation quality (PCK@0.05=0.912) and experimental results confirming sub-second latency for online tasks.
2. Furthermore, the proposed Online-MMSI-VLM demonstrates consistent improvements over baselines and even human benchmarks in certain settings.
3. The experiments are rigorous, including visual prompting and backbone scale evaluations, provides a clear and convincing validation of the model’s effectiveness.